# Visible-light-excited robust room-temperature phosphorescence of dimeric single-component luminophores in the amorphous state

Danman Guo[1], Wen Wang[1], Kaimin Zhang[1], Jinzheng Chen[2], Yuyuan Wang[1], Tianyi Wang[1], Wangmeng Hou [2], Zhen Zhang [2], Huahua Huang [2], Zhenguo Chi [1] & Zhiyong Yang [1,3] ✉

Organic room temperature phosphorescence (RTP) has significant potential in various applications of information storage, anti-counterfeiting, and bio-imaging. However, achieving robust organic RTP emission of the single-component system is challenging to overcome the restriction of the crystalline state or other rigid environments with cautious treatment. Herein, we report a single-component system with robust persistent RTP emission in various aggregated forms, such as crystal, fine powder, and even amorphous states. Our experimental data reveal that the vigorous RTP emissions rely on their tight dimers based on strong and large-overlap $\pi$-$\pi$ interactions between polycyclic aromatic hydrocarbon (PAH) groups. The dimer structure can offer not only excitons in low energy levels for visible-light excited red long-lived RTP but also suppression of the nonradiative decays even in an amorphous state for good resistance of RTP to heat (up to 70 °C) or water. Furthermore, we demonstrate the water-dispersible nanoparticle with persistent RTP over 600 nm and a lifetime of 0.22 s for visible-light excited cellular and in-vivo imaging, prepared through the common microemulsion approach without overcaution for nanocrystal formation.

Afterglow or persistent luminescence, whose emission can be clearly observed by naked eyes after removal of the excitation source, has aroused considerable attention in the fundamental study of organic excitons and a variety of application fields such as information storage[1], anti-counterfeiting[2,3] and bio-imaging[4–6]. Especially in recent decade, substantial progress has been made in pure organic room temperature phosphorescence (RTP) materials due to their unique advantages such as abundant molecular architectures, fine bio-compatibility, and good processability[7–10]. In order to achieve efficient

RTP, there are two critical issues addressed for organic materials. One is to promote the intersystem crossing (ISC) efficiency from the lowest singlet state ($S_1$) to triplet states ($T_n$) for phosphorescence[11,12]. The other is to suppress adverse deactivation and quenching processes of $T_n$[13]. Thus, for most organic luminophores, there is a prerequisite to constructing a rigid environment to stabilize triplet excitons and subsequently activate their phosphorescence at room temperature. Different strategies, including crystallization, host-guest complexation, polymerization, and matrix rigidification, have been developed to

[1]PCFM Lab, Guangdong Engineering Technology Research Center for High-performance Organic and Polymer Photoelectric Functional Films, GBRCE for Functuional Molecular Engineering, School of Chemistry, Sun Yat-sen University, Guangzhou 510275, P. R. China. [2]PCFM Lab, School of Materials Science and Engineering, Sun Yat-sen University, Guangzhou 510275, P. R. China. [3]Guangdong Provincial Key Laboratory of Optical Chemicals, XinHuaYue Group, Maoming 525000, P.R. China. ✉e-mail: yangzhy29@mail.sysu.edu.cn

reduce the nonradiative decay of $T_n$ efficiently[14,15]. However, the stringent treatment for crystal formation and inevitable phase separation in doping systems or host-guest complexes severely restricted the application and development of RTP materials. Indeed, researchers have yet to find a robust persistent RTP luminophore system, whether inorganic or organic, whose phosphorescent emission is not restricted to the crystalline state or other rigid environments with cautious treatment[16–22].

Crystallization is the simplest and most commonly used strategy to block oxygen quencher and facilitate the rigidification of small molecules, thus leading to the achievement of efficient RTP[23]. Organic RTP crystals based on single-component luminophores or host-guest systems have been frequently explored and exhibit various phosphorescence[24–27]. Their RTP emission generally diminished or even vanished as crystal destroy via mechanical grinding or melt quenching, mainly due to the loss of delicate intermolecular interactions and oxygen barrier properties. Accordingly, some interesting RTP properties sensitive to mechanical force, oxygen, or water have been developed based on these delicate interactions[28–31]. Nevertheless, we envision that if the intermolecular interaction between two molecules is strong enough to stabilize triplet excitons and partially block oxygen, the system would still exhibit persistent phosphorescence at room temperature even when its crystal structure was destroyed. To realize this vision, polycyclic aromatic hydrocarbon (PAH)-type luminophores possessing a large $\pi$-conjugate plane come into our sights[32,33]. The highly stable dimer formed in these molecular systems due to the strong $\pi$-$\pi$ interaction between PAH groups may break the tether of crystallization and be the right candidate for achieving robust RTP in an amorphous state.

Herein, robust persistent RTP emission in various states is presented, such as crystal, fine powder, amorphous state, and even in water-dispersible nanoparticles (Fig. 1). Our design is based on PAH-type luminophores, phenyl(triphenylen-2-yl)methanone compounds, in which the triphenylene core possesses a typical large $\pi$-conjugate plane to form an electronic configuration of $^3(\pi, \pi^*)$ in excited state for stabling triplet excitons. And the phenyl carbonyl unit is aimed to promote ISC efficiency, which has an electronic configuration of $^3(n, \pi^*)$ with a fast ISC rate (Fig. 1a–c)[34]. More importantly, the strong $\pi$-$\pi$ interaction between the triphenylene cores is beneficial for forming stable dimers. These compounds show highly desirable RTP properties with deep red afterglow upon excited by visible light up to 560 nm, which is also an advantage of forming dimers (Fig. 1d). Relative to ultraviolet (UV) light, visible light is a safer excitation source to animals and human beings. Visible-light excited RTP has been reported based on some metal-free organic systems (Suppl. Table 1)[4,24,35–44]. However, deep red afterglow with the maximum RTP peak of above 600 nm among them was still scarce[45]. Moreover, herein visible-light excited long-lived red phosphorescence existed not only in their crystalline states but also after mechanical ground and even in the amorphous states after melt quenching treatment (Fig. 1e). These compounds also showed the same red afterglow in water-dispersible nanoparticles prepared without overcaution for nanocrystal formation. What is important, such robust RTP emission originated from the highly stable dimer structures based on PAH-type molecules that are of significance

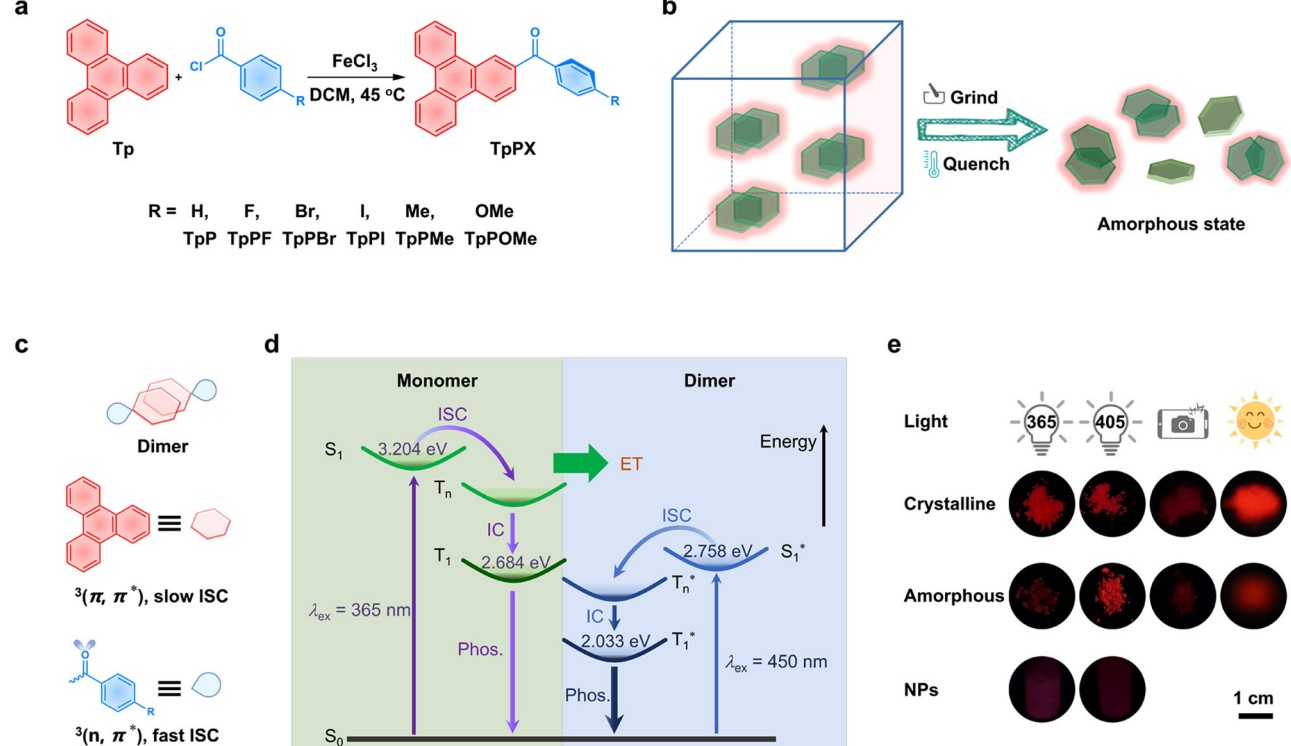

**Fig. 1 | Schematic representation for visible-light-excited robust RTP based on dimeric luminophore. a** Synthetic routes to the designed compounds. **b** Illustration of robust RTP for organic crystals based on dimeric luminophore. **c** Molecular design and the dimer formation for persistent RTP. The red plane, such as the triphenylene core, possesses an electronic configuration of $^3(\pi, \pi^*)$ in an excited state and subsequent slow intersystem crossing (ISC). The blue unit, such as phenyl carbonyl, has an electronic configuration of $^3(n, \pi^*)$ with a fast ISC rate. **d** Proposed mechanism for visible-light-excited RTP based on dimeric lumino-phore. The energy levels of $S_1$, $T_1$, $S_1^*$, and $T_1^*$ were calculated from their luminescence spectra and inset. S for singlet excited state, T for triplet excited state, without and with * referred to as monomer and dimer, respectively. ET for energy transfer and IC for internal conversion, respectively. The purple and blue arrows for the excitons transition processes in monomer and dimer for RTP, respectively. **e** Long-lived luminescent photographs of TpPBr in different states: crystalline powder, amorphous powder, and TpPBr@F127 nanoparticles. The photographs were taken after removing the excitation sources (from left to right: flashlight with $\lambda = 365$ or 405 nm, mobile phone, and solar simulator).

in promoting the formation of triplet excited states in low energy levels and suppressing the nonradiative decays even in the amorphous states.

## Results

### Characterisation of TpP and its derivatives

Phenyl(triphenylen-2-yl)methanone (TpP) and its derivatives (TpPX) were readily synthesized by a one-step reaction from triphenylene and benzoyl chloride or its derivatives (Fig. 1a). Their chemical structures with high purity were confirmed by analysis of $^1H$ and $^{13}C$ nuclear magnetic resonance (NMR), high-resolution mass spectrometer (HR-MS), and high-performance liquid chromatography (HPLC) (Suppl. Figs. 1–4). Their crystals could be easily obtained by recrystallization using dichloromethane and absolute ethyl alcohol mixed solvents, and revealed by wide-angle X-ray diffraction (XRD) results in Suppl. Fig. 5a. Their typical photoluminescence properties were summarized in Suppl. Fig. 6 and Suppl. Tables 2–4 as well.

### Robust RTP and its mechanism of dimeric TpPBr

Firstly, the RTP properties of TpPBr as a typical model were discussed in detail. Under different excitation lights from UV of 365 nm to visible of up to 560 nm, the profiles of phosphorescent spectra of TpPBr crystal kept approximately the same. The main emission was in the range of 570–750 nm with the maximum phosphorescent peak at about 600 nm, as shown in their delayed spectra measured at 8 ms

(Fig. 2a) or even 99 ms (Suppl. Fig. 7) after turning off the excitation light. The phosphorescence characteristic of TpPBr was also supported by the continuous increase in its emission intensity and lifetime with temperature decreasing (Suppl. Fig. 8 and Suppl. Table 5). From the excitation-phosphorescence mapping (Fig. 2b), it is clear that TpPBr can be efficiently excited from 390 to 560 nm, with the strongest excitation at 490 nm. After turning off the excitation light, the bright red afterglow of over 2 s can be observed by the naked eyes. Impressively, the long-lived red emission of TpPBr powder was still seen and lasted >1.5 s, after the crystal was destroyed by treatments of mechanical grinding or even melt quenching (Fig. 2c and Suppl. Movies 1, 2). As shown in the wide-angle XRD pattern of Fig. 2d, there was only a very broad peak without any sharp scattering signals, verifying its amorphous state via melt quenching. The amorphous state was further revealed by differential scanning calorimetry (DSC) measurement (Fig. 2e), where there was a cold crystallization peak for the quenching powder while there was no peak for the crystal and the ground samples.

From the analysis of their delayed luminescence spectra (Fig. 2f, g), it is found that the amorphous sample of TpPBr showed almost the same phosphorescent emission as its crystal did, using either UV or visible light excitation. Furthermore, the lifetime of persistent phosphorescence at 610 nm in the amorphous state was still over 1.6 ms in the air (Fig. 2h). Under vacuum conditions, the phosphorescent lifetime of the amorphous sample was as long as 0.23 s,

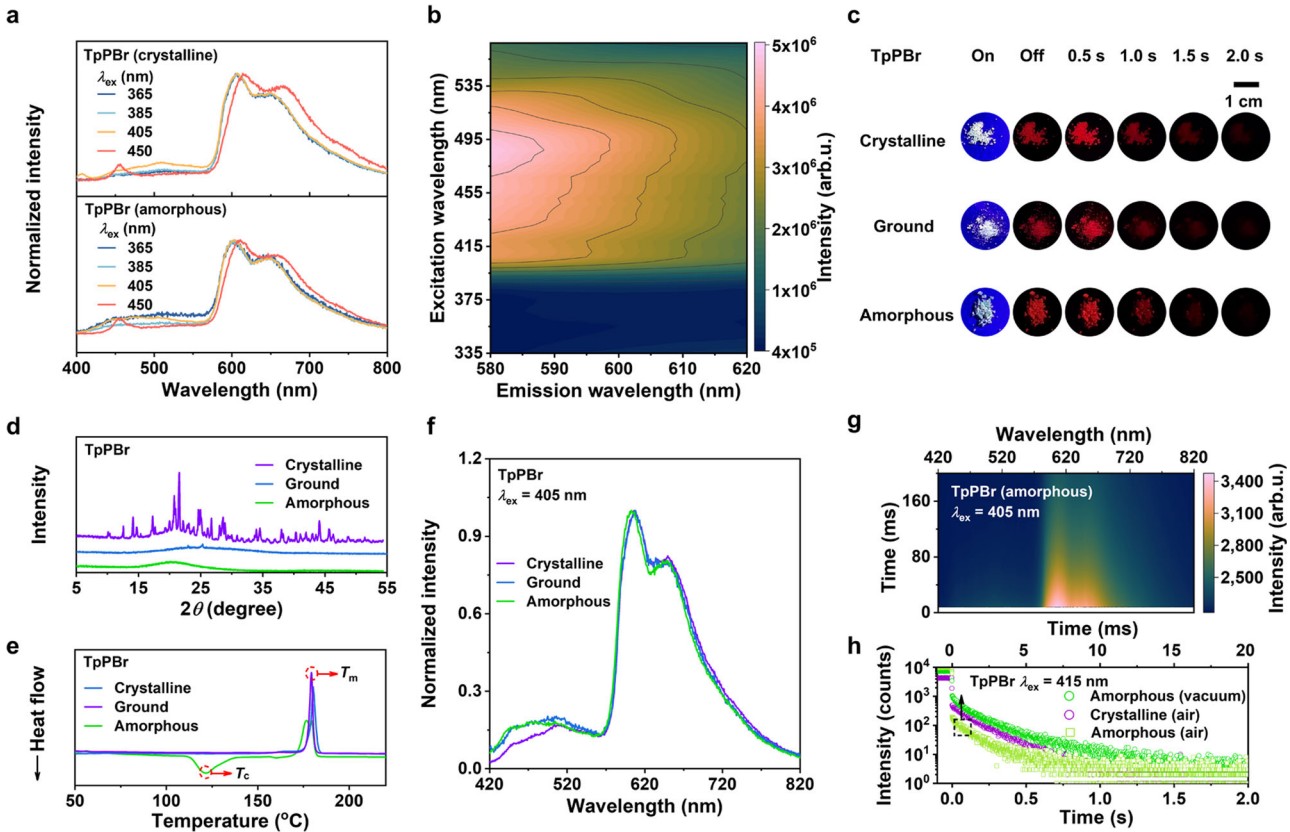

**Fig. 2 | Phosphorescent properties of typical dimeric luminophore of TpPBr with robust RTP. a** Delayed luminescence spectra of crystalline and amorphous TpPBr powder at the different excitation wavelength (298 K, in air, delayed 8 ms). **b** Excitation-phosphorescence mapping of TpPBr powder (298 K, in air). The color change from dark blue to pink represents the increase in emission intensity. **c** Luminescent photographs of crystalline, ground, and amorphous TpPBr powder before and after turning off the excited light (298 K, in air, $\lambda_{ex} = 405$ nm). **d** Wide-angle X-ray diffraction (XRD), and (**e**) differential scanning calorimetry (DSC) of crystalline, ground, and amorphous TpPBr powder. DSC data were collected from

the first heating cycle. **f** Delayed luminescence spectra of crystalline, ground, and amorphous TpPBr powder (298 K, in air, $\lambda_{ex} = 405$ nm). **g** Transient photoluminescence (PL) decay mapping of amorphous TpPBr powder. The color change from pink to dark blue represents the decrease in emission intensity (298 K, in air, $\lambda_{ex} = 405$ nm). **h** PL intensity decay curves of crystalline and amorphous TpPBr powder in a vacuum or air at the peak of 610 nm (298 K, $\lambda_{ex} = 415$ nm). The black box and arrow indicate the light green scatters corresponding to the upper horizontal axis.

being equal to the one of its crystals as shown in the photoluminescence (PL) intensity decay curves of Fig. 2h. It means that the reduced lifetime in the air of the amorphous sample is mainly due to the quenching by environmental oxygen. All these results indicate that the phosphorescence emission of both the crystal and the amorphous TpPBr samples are derived from the same luminescence center, which was highly stable and still existed even with mechanical grinding or melt quenching treatments.

For a deeper understanding of the uniquely robust RTP properties, a set of spectral studies, single-crystal XRD analysis, and theoretical calculations on the TpPBr molecule were further conducted. The prompt luminescence spectra of TpPBr in 1,4-dioxane solution with various concentrations were first studied. As shown in Fig. 3a, when the concentration increased from a highly diluted one of 0.05 mM to 1.0 mM, the luminescence intensity at 430 nm ascribing to the TpPBr monomer obviously decreased. While its content further increased up

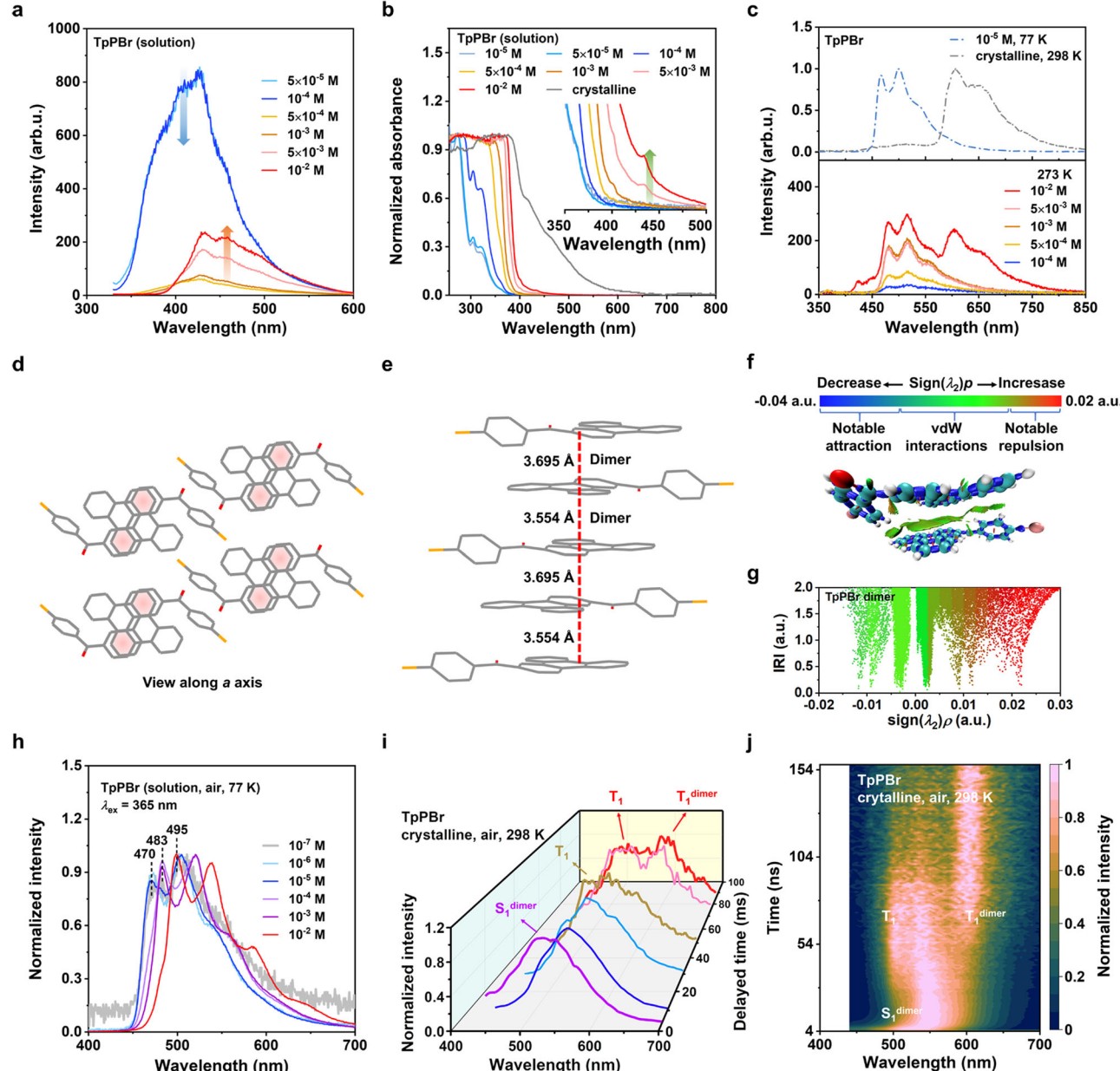

**Fig. 3 | Characterization of the formation of TpPBr dimer for its robust RTP.** **a** Prompt luminescence spectra and **b** UV-vis absorption of TpPBr in 1,4-dioxane solutions with incremental concentrations (Prompt: 298 K, in air, $\lambda_{ex}$ = 310 nm; UV-vis: 298 K, in air). The blue arrow indicates the decrease in emission intensity around 430 nm, while the red arrow for the increase around 460 nm in 3a. The green arrow in 3b indicates the appearance of the absorption peak at about 440 nm. **c**, Normalized delayed luminescence spectra of TpPBr in 1,4-dioxane (0.01 mM) measured at 77 K and TpPBr crystalline powder measured at 298 K in air (upper); Delayed luminescence spectra of TpPBr in 1,4-dioxane solutions after deoxygenation with incremental concentrations measured at 273 K (lower, $\lambda_{ex}$ = 365 nm, delayed 8 ms). The packing structure of TpPBr dimer in its crystal

structure, viewed along *a* axis (**d**) and another non-axial angle (**e**). **f** Distribution regions and (**g**) scatter plots of intramolecular/intermolecular interactions in TpPBr dimers revealed by the interaction region indicator (IRI), a natural space function. The sign($\lambda_2$)*p* function is coloured on IRI isosurfaces to illustrate the nature of interaction regions. A decrease in sign($\lambda_2$)*p* is indicated by the colour bar changing from red to green and then to blue, which implies a notable repulsion, a noticeable interaction, and a significant attractive effect, respectively. vDW for van der Waals. **h** Delayed luminescence spectra of TpPBr in 1,4-dioxane solutions with incremental concentrations measured at 77 K ($\lambda_{ex}$ = 365 nm, delayed 8 ms). **i,j** Normalized time-resolved emission spectroscopy (TRES) of TpPBr powder in air at 298 K in nanosecond scale ($\lambda_{ex}$ = 405 nm).

to 10.0 mM, the intensity at about 460 nm was significantly enhanced, implying the formation of an aggregated luminescence center of TpPBr molecules. The relative fluorescence lifetime at about 460 nm increased first and then decreased along with the concentration change (Suppl. Fig. 9). It also revealed the dimers/excimers formed and subsequently aggregated in the solution. Under the same concentration of 10.0 mM, it was found that a small absorption peak clearly appeared at around 440 nm in the UV-visible absorption spectra of TpPBr in 1,4-dioxane solution (Fig. 3b). These results indicate that the TpPBr aggregated into dimers not excimers in a high concentration of 10.0 mM[46]. Additionally, a sole delayed luminescence at 400–550 nm was obtained in the highly concentrated solution at 77 K (Fig. 3h). This luminescence should belong to phosphorescence from monomers but not mixed multimers, as they were neither broad nor multiple mixed peaks along aggregates forming in the solution. Furthermore, the delayed luminescence spectra of TpPBr in 1,4-dioxane solution at 0 °C can be measured at 8 ms delayed after turning off the excited UV light. In the literature, the phosphorescent test in a solid solution was generally carried out at a very low temperature of 77 K, which, however, suffered a significant influence of temperature on the emission of the phosphors. Herein, we skillfully used 1,4-dioxane possessing a high melt temperature of 12 °C as a solvent, so the molecular motion of TpPBr in its solution at 0 °C can be efficiently inhibited. Thus, the phosphorescent emission of TpPBr can be characterized in a solid solution of 1,4-dioxane at 0 °C to avoid the disturbance of low temperature. As shown in Fig. 3c, within the concentration range from 0.1 mM to 5.0 mM, their delayed luminescence spectra exhibited almost the same with the emission in the 450–580 nm range, ascribing to the dispersed TpPBr molecules. When the concentration was as high as 10.0 mM, the delayed luminescence spectrum showed new emission peaks in the long-wavelength range of 600 - 750 nm with a maximum peak of about 600 nm, which should be derived from the TpPBr dimers. Meanwhile, the profile of these long-wavelength emission peaks was nearly identical to the prominent phosphorescent peaks of the crystalline and amorphous TpPBr samples. The time-resolved spectral decay of TpPBr crystalline powder in different conditions of atmospheres and temperatures further confirmed this mechanism, as shown in Fig. 3i, j and Suppl. Fig. 10. Thus, these results demonstrate that the red phosphorescent emission with the maximum peak at about 610 nm in the crystalline and the amorphous TpPBr samples originated from the dimers. It also revealed the high stability of the TpPBr dimers, which can exhibit phosphorescence not only in the crystal but also in the amorphous state.

To take a closer look at the molecular packing of the dimer, single-crystal XRD of TpPBr (Fig. 3d, e) revealed strong intermolecular $\pi$-$\pi$ interactions between the triphenylene units, with distances of 3.554 and 3.695 Å, respectively. Correspondingly, two types of dimers of TpPBr could be formed between the molecular packing structures, which were analyzed as H- and J- aggregation using the molecular exciton theory[47], respectively (Suppl. Figs. 11, 12 and Table 7). Additionally, there is a significant overlap of a biphenyl structure between the TpPBr molecules in the dimer structure, as clearly seen along $a$ axis of the single crystal and another non-axial angle (Fig. 3d, e). Furthermore, a natural space function named interaction region indicator (IRI) was used to reveal interactions of molecules, which can simultaneously show covalent and noncovalent interactions in a single map. As shown in the isosurface map of IRI = 1.1, it can be seen that a very broad green isosurface with sign($\lambda_2$)$\rho$ value <−0.005 exists in the middle of two triphenylene units in the dimer structure, revealing the van der Waals (vdW) attractive interaction between the two molecules of TpPBr (Fig. 3f, g and Suppl. Fig. 13). In addition, there are several orange spots on this green isosurface, which indicates the presence of weak repulsion between the two triphenylene planes due to the crowded $\pi$ orbitals. As a result, the intermolecular $\pi$-$\pi$ interaction inside the TpPBr dimer is relatively strong to keep in various aggregated states,

even in an amorphous state. The calculated data also indicated that hole/electron located on both molecules in the dimer structure and a larger spin-orbit coupling (SOC) of TpPX for promoting phosphorescence (Suppl. Figs. 14, 15). The experimental quantum yield of ISC ($\Phi_{ISC}$) and phosphoresce rate constant ($k_P$) of TpPX were increased when compared to the Tp core itself (Suppl. Table 3). These results illustrate the crucial roles of the triphenylene unit in forming the stable dimers and the carbonyl group in promoting ISC to achieve such robust and highly desirable RTP properties in different states of TpPBr.

## Triplet exciton stabilization through PAH-based dimer

As we know that visible-light excited organic RTP materials with red afterglow were scarce and yet to be reported for a single-component amorphous system. To demonstrate the general application of triplet exciton stabilization through the dimer strategy based on PAH groups for achieving robust persistent RTP, TpP derivatives with different substituent groups, TpPX, were designed and synthesized. In addition, it is valuable to understand the influence of substituent groups on the RTP property of amorphous TpP derivatives (Suppl. Fig. 6 and Suppl. Tables 2–4). All the amorphous TpPX samples were obtained via melt quenching, confirmed by the XRD analysis (Suppl. Fig. 5b). Two visible light sources, including 405 and 450 nm, were used as the excitation, respectively. As shown in their delayed luminescence spectra of Fig. 4a, b measured in the atmosphere environment, the TpPI compound containing iodine substituent exhibited the strongest phosphorescence intensity among six TpPX compounds under the excitation with 405 or 450 nm light source. This is reasonable that the significant heavy-atom effect of iodine is beneficial to SOC, leading to efficient ISC for promoting RTP. On the other hand, the iodine atom possesses a large atomic radius of 1.33 Å, which may partially exhibit an intermolecular SOC effect and further enhance the phosphorescence. By comparison, it was found that the emission intensity at about 600 nm followed the order: TpPI > TpPBr > TpP ≈ TpPF ≈ TpPMe > TpPOMe, regardless of using UV (Suppl. Fig. 16) or visible light (Fig. 4a, b) for excitation. Similarly, the profiles of the delayed spectra of these amorphous samples were almost the same as those of their crystals, especially in the red emission range (Suppl. Fig. 17). More interestingly, the emission intensities of these TpPX compounds measured under vacuum conditions were slightly stronger than those in the atmosphere environment, as shown in Fig. 4c and Suppl. Fig. 18. These results mean that these amorphous structures can efficiently stabilize triplet excitons and suppress oxygen quenching through the tight dimers inside. As revealed through single-crystal XRD of six TpPX compounds and Tp, their tight dimer structures were confirmed and ascribed as H- or J-aggregates by analyzing their molecular packing structures with the distance of triphenylene plane in the range of 3.374–3.695 Å (Suppl. Tables 6, 7 and Suppl. Figs. 11, 12)[47]. The Tp core itself showed an H-aggregation in the crystal. After being substituted by the phenyl carbonyl unit, only TpPI and TpPOMe adopted H-aggregation in their two dimer types in the crystals. The dimers of TpP and TpPF belonged to J-aggregates. Regarding TpPBr and TpPMe, they contained two types of dimers and formed mixed H/J-aggregates in their crystals. Additionally, similar strong intermolecular $\pi$-$\pi$ interactions inside the dimer of TpPX are also found in the natural transition orbital and the isosurface map of IRI (Suppl. Figs. 13, 14).

From the time-resolved emission decay curves of these amorphous samples under the visible-light excitation of 405 nm (Fig. 4d), the lifetimes at 630–655 nm of four compounds, including TpP, TpPF, TpPBr, and TpPMe were at a millisecond level of 1.4–5.3 ms, while the lifetime of TpPI was the longest one of up to 120 ms. When measured in vacuum, the lifetimes of all these amorphous compounds increased (Fig. 4e). In particular, the lifetimes of TpPBr and TpPMe in amorphous states measured in vacuum, were almost close to the ones of their crystals measured at ambient conditions, respectively. The TpPI amorphous sample measured in vacuum even had a longer

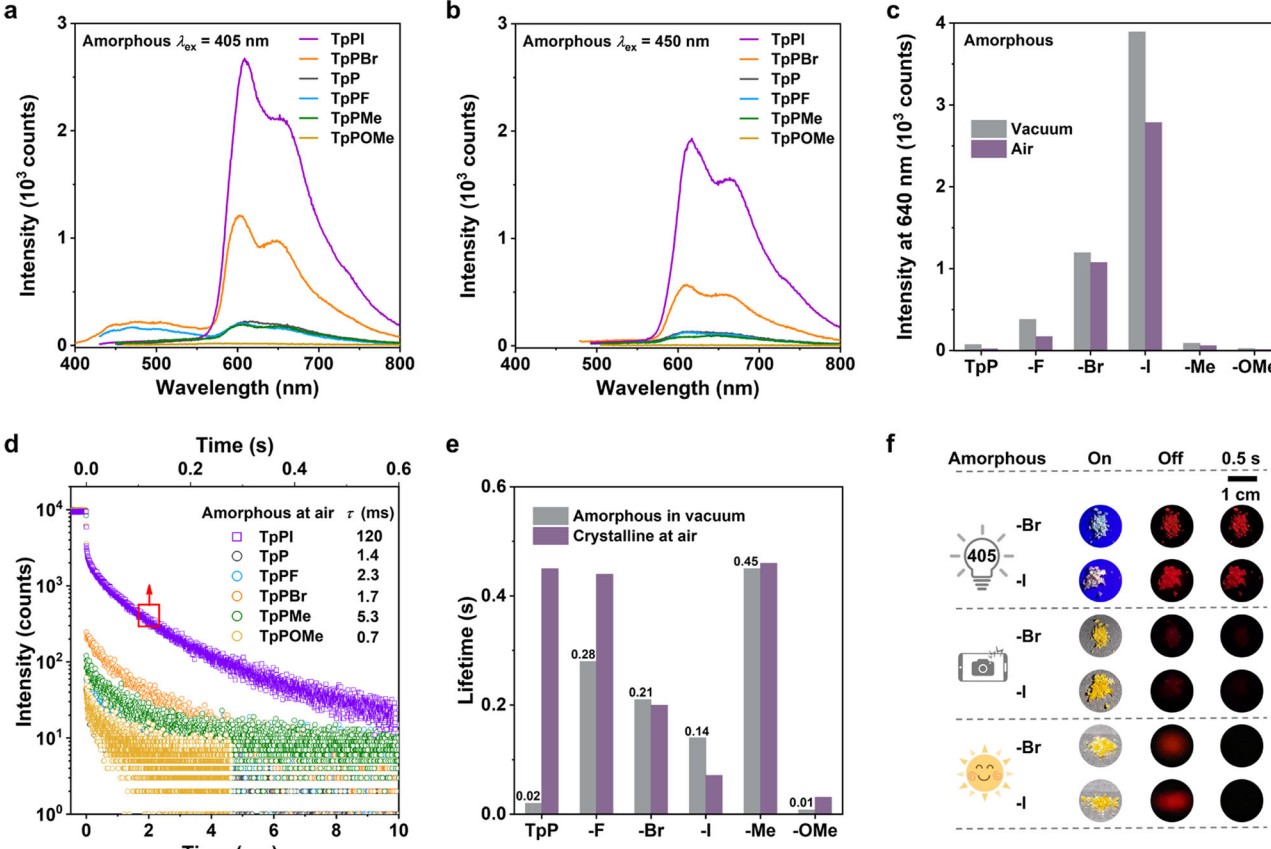

**Fig. 4 | Phosphorescent properties of dimeric luminophores of TpPX.**
**a**, **b** Delayed luminescence spectra of amorphous TpPX powder under excitation at 405 nm (**a**) and 450 nm (**b**) (298 K, in air, delayed 8 ms). **c** The phosphorescence intensity of amorphous TpPX powder in the vacuum or air conditions at the peak of 640 nm (298 K, $\lambda_{ex}$ = 405 nm). **d** Photoluminescence intensity decay curves of amorphous TpPX powder at its corresponding phosphorescent peak (298 K, in air, $\lambda_{ex}$ = 415 nm). **e** Comparison between the lifetime of TpPX powder in the crystalline (in the air) and the amorphous (in the vacuum) states ($\lambda_{ex}$ = 415 nm). **f** Luminescent photographs of amorphous TpPX powder after turning off the excited light (flashlight with $\lambda_{ex}$ = 405 nm, mobile phone, and solar simulator).

phosphorescence lifetime than its crystal did. This reflected that the TpPI crystal structure was not close-packed enough, mainly due to its "head-to-head" parallel packing and the steric hindrance of the iodine atom with a large atomic radius. Taken together, because of the iodine and bromine atoms with the relatively efficient heavy-atom effect, both amorphous samples of TpPBr and TpPI exhibit relatively strong phosphorescence emission with bright red afterglow up to over 1.5 s, which can be observed by the naked eyes after turning off the excited light of 405 nm (Fig. 4f and Suppl. Movie 3). Moreover, even when using other safer light sources such as mobile phones and sunlight, their red afterglow can also be excited, because of their broad absorptions extending to 560 nm (Suppl. Fig. 19).

**Application of robust RTP of PAH-type materials**
To demonstrate the stability of the unusually robust RTP for the application potential of this kind of PAH-type materials, the RTP emission of TpPBr crystal and its amorphous sample was first measured under different heating conditions. As shown in Fig. 5a, b, the phosphorescence of both samples can be clearly detected even when the crystalline and amorphous samples were heated up to 80 and 70 °C, respectively; meanwhile, their red afterglow could still be captured by the camera (Fig. 5c). In addition, as displayed in Fig. 5d, e, when the water was dropped onto TpPBr amorphous sample, its bright long-lived emission up to 3.0 s was observed clearly, which was similar to that measured without water (Fig. 4f). These results further reflect that RTP emission of TpPBr sample has the strong resistance to high temperature and water (or moisture), attributing to its tight dimer structure.

In addition, the visible-light excited red afterglow with a maximum emission peak of 600 nm is highly beneficial for application in bio-imaging. However, in the literature, it is essential to construct organic nanocrystals to retain the afterglow, which is only emitted in the crystalline state[4]. Fortunately, this desirable RTP property can be achieved even in the amorphous form of these PAH-type samples. This means that overcaution treatments for preparing their nanocrystals are not necessary yet. Therefore, the TpPBr nanoparticle with persistent RTP was easily prepared through the common microemulsion approach by using dichloromethane as the oil phase and Pluronic F127 as the surfactant. The prepared nanoparticle has a uniform size of about 370 nm, measured by dynamic light scattering (DLS). Also, the amorphous form containing a small minority of TpPBr microcrystal mainly existed in the nanoparticle, which was revealed by transmission electron microscope (TEM) and selected area electron diffraction (SAED) techniques (Fig. 5f). To our delight, the phosphorescence emission of TpPBr nanoparticle was still detected even when the concentration of TpPBr was as low as 2.4 mM (Fig. 5h, i and Suppl. Fig. 20). Moreover, the red afterglow of TpPBr nanoparticle was clearly seen in the aqueous solution, and the lifetimes at 610 nm and 645 nm of the nanoparticle were as long as 0.22 and 0.31 s, respectively (Fig. 5i and Suppl. Movie 4). Such a long-lived luminescence at a long-wavelength range of over 600 nm, which can be excited by visible light over 450 nm, was rarely reported for purely organic nanoparticles (Fig. 5g, Suppl. Fig. 21, and Suppl. Table 8), which is an ideal reagent for bio-imaging. Firstly, the TpPBr nanoparticles showed excellent biocompatibility based on MTT assays in Suppl. Fig. 22. Additionally, as

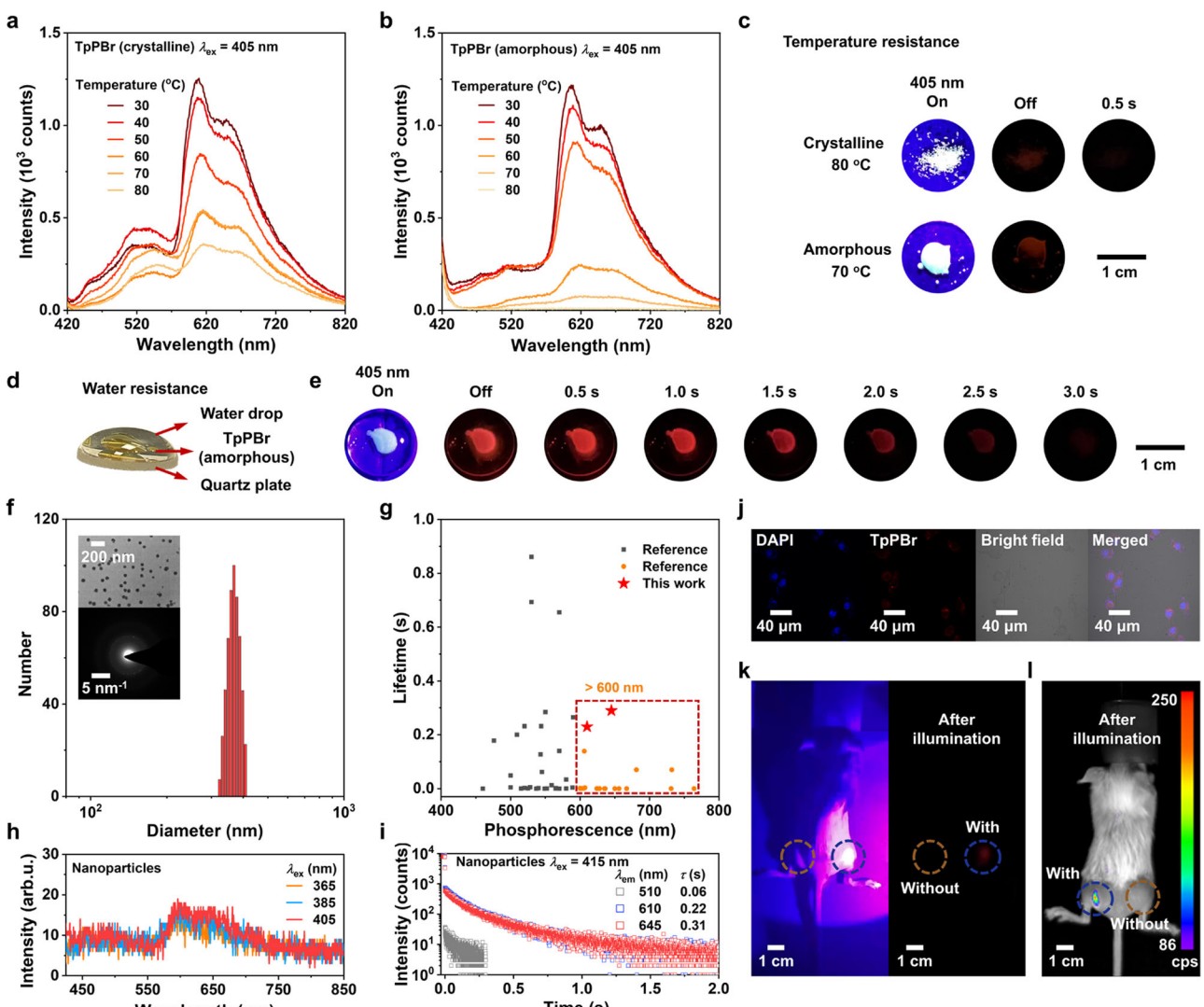

**Fig. 5 | Demonstration of the resistance to heat or water and the bio-imaging application potential of visible-light excited robust RTP of TpPBr. a, b** Delayed luminescence spectra of crystalline (**a**) and amorphous (**b**) TpPBr sample measured various temperatures from 30 to 80 °C in the air ($\lambda_{ex} = 405$ nm, delayed 8 ms). **c** Luminescent photographs of crystalline and amorphous TpPBr samples on a hot plate at 80 °C and 70 °C, respectively, captured after turning off the excited light ($\lambda_{ex} = 405$ nm). **d** An illustration of an amorphous TpPBr sample with water drop on its surface. **e** Luminescent photographs of an amorphous TpPBr sample affected by water were captured after turning off the excited light ($\lambda_{ex} = 405$ nm). **f** Size distribution of TpPBr@F127 nanoparticles. The transmission electron microscopy (TEM) and selected area electron diffraction (SAED) images were inset (bar: 200 nm). TEM and SAED experiments were repeated eight times. **g** Comparison of room-temperature phosphorescence and corresponding lifetimes of pure organic nanoparticles reported since 2016 (Suppl. refs. 6,7,9,10,14–41). The chromophores

in these nanoparticles are single-component. **h** Delayed luminescence spectra of TpPBr@F127 nanoparticles in $H_2O$ (9.76 mM) at the different excitation wavelength (298 K, in air, delayed 8 ms). **i** Photoluminescence intensity decay curves of TpPBr@F127 nanoparticles in $H_2O$ (2.44 mM, 298 K, in air, $\lambda_{ex} = 415$ nm). **j** Confocal microscopy images of RM-1 cells co-incubation with TpPBr@F127 nanoparticles. Cells were viewed in the blue channel for DAPI ($\lambda_{ex} = 405$ nm, $\lambda_{em} = 430$–470 nm), a red channel for TpPBr ($\lambda_{ex} = 488$ nm, $\lambda_{em} = 600$–650 nm), bright field and merged images, respectively. The experiment was repeated three times. **k, l** Visible-light excited afterglow in-vivo bioimaging photographs of living mice captured (**k**) before/after removing the flashlight and (**l**) an IVIS instrument in bioluminescent mode after removing the flashlight ($\lambda_{ex} = 405$ nm, 35 W m⁻²). TpPBr@F127 nanoparticles (with, blue dashed circle) were injected into one upper hind leg while F127 nanoparticles without TpPBr (without, orange dashed circle) were injected into the other as a reference.

clearly shown in the images of Fig. 5j–l, the TpPBr nanoparticles were utilized for the imaging of RM-1 cells and living mice. A bright phosphorescence emission in the red channel was collected for the cells incubated with the nanoparticles. After illumination, afterglow bioimaging photographs with autofluorescence-free for living mice were successfully captured by an IVIS instrument in bioluminescent mode or a camera with video parameters of 1080 p HD/60 fps in an ordinary accessible mobile phone (Fig. 5k, l). These results demonstrate the broad prospects for biological applications of these organic nanoparticles, which can work without real-time external excitation and avoid the interference of spontaneous fluorescence in organisms.

## Discussion

In conclusion, we present single-component organic luminophores featuring robust persistent RTP emission in various aggregated forms. Unlike common organic RTP molecules, these compounds, namely TpPX, exhibit almost the same RTP emission when they are crystals, fine powders, and even in amorphous states. Additionally, their vigorous RTP is in the spectral range over 600 nm, which can be excited by visible light as long as 550 nm as well. Both the experimental results and simulation revealed that their tight dimers formed through the strong and large-overlap π-π interactions between polycyclic aromatic hydrocarbon (PAH) groups were responsible for their vigorous RTP

emissions. The dimer structure can offer efficient suppression of the nonradiative decay even in an amorphous state for good resistance of RTP to heat (up to 70 °C) or water. The excitons of such tight dimers possess low energy levels, which promote the red-shift of both absorption/excitation and RTP emission spectra of these PAH derivatives into visible and deep-red ranges, respectively. By tailoring their molecular structures with various substituents, both amorphous samples of TpPBr and TpPI exhibit relatively intense phosphorescence with bright red afterglow up to over 1.5 s. Their lifetimes measured in vacuum were almost close to the ones of their crystals measured at ambient conditions. In particular, the lifetime at 630–655 nm of the amorphous TpPI sample measured in the air was still long and up to 120 ms under the visible-light excitation. Furthermore, through the common microemulsion approach without overcaution for nanocrystal formation, we demonstrate the application of water-dispersible TpPBr nanoparticles with persistent RTP over 600 nm and a lifetime of 0.22 s for precise cellular and in-vivo imaging. Importantly, our findings not only provide a design strategy for realizing single-component luminescent materials with robust phosphorescence in an amorphous state but also open the opportunity to develop RTP nanoparticles without the assistance of nanocrystal formation.

## Methods
### Materials
All reagents and materials are used directly without special instructions. Triphenylene (98%) was purchased from Shanghai Yuanye Bio-Technology Co., Ltd. 4-bromobenzoyl chloride (98%) was purchased from J&K Scientific Ltd. 4-iodobenzoyl chloride (98%), 1,4-dioxane (99.9%) and poly(ethylene glycol)-block-poly(-propylene glycol)-block-poly(ethylene glycol) (F127, average $M_n \approx 12,600$ g mol$^{-1}$) were purchased from Shanghai Aladdin Biochemical Technology Co., Ltd. Benzoyl chloride (99%), 4-fluorobenzoyl chloride (98%) and 4-methoxybenzoyl chloride (97%) were purchased from Shanghai Energy Chemical Co., Ltd. 4-toluoyl chloride (97%) was purchased from Alfa Aesar Co., Ltd. Ferrous chloride (99%) was purchased from Shanghai Macklin Biochemical Co., Ltd. Dichloromethane (DCM), n-hexane (n-Hex), tetrahydrofuran (THF), anhydrous ethanol absolute (EtOH) and hydrochloric acid (HCl) (99.5%) were purchased from Guangzhou Chemical Reagent Factory. Fetal bovine serum (FBS), phosphate buffer solution (PBS), basal medium, and 3-(4,5-dimethythiazol-2-yl)-2,5-diphenyltetrazolium bromide (MTT) were purchased from Thermo Fisher Scientific Co., Ltd. RM-1 cells were obtained from the Experimental Animal Center of Sun Yat-Sen University (Guangzhou, China). All animal studies were approved by the Institutional Animal Care and Use Committee (IACUC), Sun Yat-Sen University.

### Synthesis of TpPX
Triphenylene (1.00 mmol) and benzoyl chloride (or other derivatives, 1.50 mmol) were dissolved in 100 mL of DCM. Then ferrous chloride (3.00 mmol) was slowly added, and the mixture was stirred under the atmosphere and heated at 45 °C for 24 h. The reaction solution was washed with dilute hydrochloric acid and deionized water 3 times, respectively. Then the crude product was purified by silica gel column chromatography (DCM/n-Hex = 1/3, v/v). Finally, recrystallization by using DCM and EtOH was carried out to obtain the pristine crystalline powder. The yield is essentially 76%.

### Preparation of amorphous samples
The sample was spread on the bottom of the glass bottle. Then, it was heated to a molten state according to its $T_m$, and liquid nitrogen was quickly poured into the glass bottle. The samples were confirmed to be in an amorphous state by XRD results. For the intensity comparison between delayed luminescence spectra in a vacuum and in air, TpPX powder was prepared through heating to a molten state at a quartz

plate and subsequently quenching with liquid nitrogen. And the sample is fixed on a quartz sheet like amber.

### Preparation of nanoparticles
500 μL of TpPBr (10 mg mL$^{-1}$, DCM) was added into 5 mL aqueous solution containing F127 (10 mg mL$^{-1}$). The mixture was then sonicated by a cell crusher (sonic) for 30 s. The Microemulsion was stirred under the atmosphere and heated at 30 °C for 10 h. The mean diameter measured by DLS was 370 nm and the zeta potential was -38.69 mV.

### Cell culture and imaging
Mouse prostate cancer RM-1 cells were cultivated with a Gibco's 1640 medium that consisted of 10% fetal bovine serum (FBS) and 1% penicillin-streptomycin (PS) at 37 °C with 5% CO$_2$ and 95% relative humidity. The cells were seeded in a confocal dish and incubated with TpPBr@F127 nanoparticles (24.4 mM) in 1640 medium for 1 day. Then cells were washed twice with PBS buffer solution. Confocal microscopy imaging was performed on RM-1 cells in an FBS medium.

### Vitro cytotoxicity
To study the toxicity of nanoparticles in RM-1 cells, a 3-(4,5-dimethy-thiazol-2-yl)-2,5-diphenyltetrazolium bromide (MTT) assay was employed. The cells were cultivated into 96-well cell plates with a population of 10,000 cells per well and incubated for 24 h. Then the cells were treated with fresh medium containing free insulin and tested nanoparticles with a concentration of 2.5–1000 μM. After a 24 h incubation, the cells were washed with PBS buffer solution. And MTT solution in culture medium (0.5 mg mL$^{-1}$) was added to each well, and then the mixture was incubated for another 3 h. The cell media were removed and replaced with 100 μL of DMSO, and the absorbance was determined at 570 nm using a microplate reader. The reported cell viability is the ratio of the MTT absorbance of each well to that of the control well.

### Vivo imaging
BALB/c and C57BL/6 mouse strains were used. PBS solution of TpPBr@F127 nanoparticles (20 mg mL$^{-1}$) and F127 were injected subcutaneously into the upper hind leg of the mice, respectively. After 30 s of illumination with a flashlight ($\lambda_{ex}$ = 405 nm, 35 W m$^{-2}$), afterglow photographs of the living mice were then captured on IVIS Spectrum and Apple iPhone 12 camera with recording parameters of 1080 p HD/60 fps.

### Instrumentation and test methods
Proton nuclear magnetic resonance spectrum ($^1$H NMR) and carbon spectrum ($^{13}$C NMR) were measured on a nuclear magnetic resonance spectrometer (Brucker Avance III 500HD). The solvent was deuterated chloroform (CDCl$_3$) or deuterated dichloromethane (DCM-$d_2$), and the internal standard was tetramethylsilane (TMS). High-resolution mass spectrum was obtained with a Bruker ultrafleXtreme matrix-assisted laser desorption/ionization time of flight mass spectrometry (MALDI-TOF MS). Purity Analysis was performed on a high-performance liquid chromatograph (HPLC, Agilent 1290 Infinity). Wide-angle X-ray diffraction (XRD) was detected on a RIGAKU X-ray powder diffraction (D-max 2200 VPC). The X-ray source was Cu Target at 40 kV and 26 mA, and the scanning speed was 10° min$^{-1}$. Single crystal data of Tp, TpP, TpPF, TpPBr, and TpPI were obtained from Rigaku X-ray single crystal diffractometer (Supernova) with Cu-Kα radiation ($\lambda$ = 1.54184 or 1.34138 Å) while data of TpPOMe were obtained from Bruker D8 Ventue Photon III with Ga-Kα radiation ($\lambda$ = 1.34138 Å). The structures were solved by direct methods following the difference Fourier syntheses, and refined against all data using the SHELXTL software package as implemented in Olex2. Weighted R factors ($R_w$) and all goodness of fit $S$ are based on $F^2$; conventional $R$ factors ($R$) were based on $F$. All non-hydrogen atoms were refined with anisotropic thermal parameters.

The hydrogen atom positions were calculated geometrically and were allowed to ride on their parent carbon atoms with fixed isotropic $U$. Summary diffraction and refinement statistics can be found in Suppl. Table 6. Differential scanning calorimetry (DSC) was carried out on a differential scanning calorimeter (Netzsch DSC 204 F1) with a heating rate of 20 °C min$^{-1}$ (atmosphere: nitrogen, 20 mL min$^{-1}$). The ultraviolet-visible absorption (UV-vis) spectrum of the solution and UV-vis of the powder were carried out by Hitachi U-3900 spectrophotometer and Shimadzu UV-3600, respectively. The prompt luminescence spectrum was measured on a fluorescence spectrometer (Shimadzu RF-5301PC) or a Horiba JY FL-3 steady-state/transient combined fluorescence spectrometer or an Edinburgh FLS1000 steady/transient state fluorescence spectrometer with a Xenon light source. Delayed luminescence spectrum was performed on a miniature optical fiber spectrometer (Ocean Optics QE65PRO) with the LED light source or Edinburgh FLS100 steady-state/transient fluorescence spectrometer with a microsecond flashlamp (uF2). Photoluminescence (PL) decay profiles were collected by a Horiba JY FL-3 steady-state/transient combined fluorescence spectrometer or an Edinburgh FLS1000 steady-state/transient fluorescence spectrometer. Photoluminescence quantum yields (PLQY) were determined on an Edinburgh FLS1000 with a Xenon light source with a calibrated integrating sphere. Delayed luminescence spectra of solution at low temperatures were measured by freezing it with liquid nitrogen. Variable temperature measurements and vacuum experiments were controlled by an cryometer (Oxford OPTIDN2). For the intensity comparison between the delayed luminescence spectrum of amorphous film in a vacuum and in air, the amorphous film was put into a quartz bottle and evacuated until the pressure in the quartz bottle reached -900 mbar. The solar simulator was a 500 W Xenon Lamp Solar Simulator (Gloria-X500A). Unless otherwise specified, the spectroscopic tests were performed at room temperature and atmospheric atmosphere by default. Dynamic light scattering (DLS) and zeta potential were applied using a Brookhaven EliteSizer. The morphology of nanoparticles was assessed by transmission electron microscopy (TEM) on FEI Tecnai 12 (USA) and Hitachi HT7800. Confocal laser scanning microscopy (CLSM) images were obtained from Olympus FV3000. Afterglow photographs of the living mice were then captured on IVIS Spectrum (NightOWL II LB983, filter: 620 nm, exposure time: 30 s) instantly after irradiation of the mice with the flashlight ($\lambda_{ex} = 405$ nm, 35 W m$^{-2}$). Luminescent photographs of crystalline, ground, and amorphous compound samples before and after turning off the flashlight ($\lambda_{ex} = 365$ or 405 nm, Apple iPhone 7 Plus) were captured by an Apple iPhone 12 camera with recording parameters of 1080p HD/60 fps under air atmosphere. And ones of compound before and after turning off the solar simulator were collected by a digital camera (Canon EOS 750D). Due to the difficulty in distinguishing the brightness of the afterglow generated by the solar simulator from the ambient light, the method of capturing the afterglow was to adjust the camera exposure to the highest. Therefore, when the solar simulator was not turned off, the image captured by the camera was all white. When the light is turned off, a clear afterglow can be captured. To present the appearance of the sample in a solar simulator, the "on" image in Fig. 4f was taken normally with an Apple iPhone 12 camera with recording parameters of 1080p HD/60 fps.

### Theoretical methods
The single molecule simulation calculations were based on the single crystal conformation using the Gaussian 09 W package, time-dependent density functional theory (TDDFT) based on the B3LYP/6-31 G(d) method. The dimers were calculated with M062X/6-31 G(d) based on the single crystal conformation to take into consideration the weak long-range intermolecular interactions. Due to the lack of definition for iodine atoms in 6-31 G(d), the single molecule and dimers of TpPX were calculated using B3LYP/def2SVP and M062X/def2SVP,

respectively. Natural transition orbitals (NTOs), and interaction region indicators (IRI) were obtained by Multiwfn and visualized using VMD[48,49]. Spin-orbit coupling matrix elements (SOC) were calculated at the TDDFT by orca based on the M062X TZVP basis set, only triplet excited states not exceeding 0.5 eV above or 1.5 eV below the energy level of $S_1$ were shown in the SOC diagrams. The transition electric dipole moments of $S_0$-$S_1$ were calculated using M062X/def2SVP for TpPI and M062x/6-31 G(d) for the other compounds, including the "nosymm" keyword. This ensures that the position and orientation of the calculated system match the input file, making it easier to determine the orientation of the monomeric transition dipole moment in the dimeric coordinate system. With the help of dummy atoms positioned at the centers of the two monomers in the dimer in GaussView, the angle between the transition moment of the monomer and the interconnection of the centers was obtained, and this angle is defined as $\theta$. This simplified calculation is only valid for compounds in a co-planar inclined model. When $\theta$ is >54.7°, H-aggregation forms with positive exciton splitting energy, whereas $\theta$ < 54.7° results in J-aggregation.

### Statistics and reproducibility
The TEM and SAED experiments were repeated at least eight times. Confocal microscopy images were repeated three times. Biological experiments were repeated at least three times.

### Reporting summary
Further information on research design is available in the Nature Portfolio Reporting Summary linked to this article.

### Data availability
The data that support the findings of this study are available from the corresponding author upon request. Crystallographic data have been deposited at the Cambridge Crystallographic Data Centre (CCDC), under deposition numbers CCDC 2304917 (Tp), 2276028 (TpP), 2276075 (TpPF), 2276105 (TpPBr), 2276126 (TpPI), 2276143 (TpPMe), and 2304903 (TpPOMe).

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

## Acknowledgements
We gratefully acknowledge the financial support from the Natural Science Foundation of China (No. 52373199 (Z. Y.), 52073315 (Z. Y.), 51873237 (Z. Y.), and 51973239 (Z. C.)), the Natural Science Foundation of Guangdong (No. 2023A1515012679 (Z. Y.)), the Guangdong Natural Science Funds for Distinguished Young Scholar (No. 2017B030306012 (Z. Y.)), and the Fundamental Research Funds for the Central Universities (23yxqntd002, Z. C.).

## Author contributions

D. Guo, H. Huang and Z. Yang conceived the investigation. D. Guo, W. Wang, K. Zhang and J. Chen performed the experiments. D. Guo, Y. Wang, T. Wang, W. Hou and Z. Zhang were involved in the analyses and interpretation of data. D. Guo wrote the manuscript with the help of H. Huang, Z. Chi and Z. Yang.

## Competing interests

The authors declare no competing interests.
