## [Peer Review File · Nature Communications]

Visible-light-excited robust room-temperature phosphorescence of dimeric single-component luminophores in the amorphous stateREVIEWER COMMENTS

Reviewer #1 (Remarks to the Author):

In the manuscript entitled "Title: Visible-light-excited robust room temperature phosphorescence derived from a single-component dimer luminophore in the amorphous state" authors have designed a molecular system using triphenylene that can exhibit phosphorescence in different aggregated forms like crystal, amorphous fine powder and water-dispersible nanoparticles using a single component. Authors have performed extensive investigation to reveal the superior stability of the system towards mechanical grinding, which was shown by the unchanged emission intensities and long-lifetimes of the powder after melt quenching experiments. The present system is also highly resistant towards temperature and water, as clear from the efficient phosphorescence and long afterglow even in the presence of water and at high temperature. The manuscript has enough novelty and suitable to publish in this journal. However, the author should address the following points.

1. Since the system exhibits quite efficient afterglow, authors can think about adding the term 'Afterglow' in the title to elevate the impact on readers.
2. The writings on schematics of Figure 1b are not visible, please change them.
3. What does the feature from 500-560 nm in Figure S7 signify, monomeric fluorescence or phosphorescence. And, why does the change for this band is irregular in temperature dependent emission spectra.
4. Authors should also record lifetime of the band before the phosphorescence band from the dimer to understand its origin unambiguously.
5. Although, there is a sustenance on the type of aggregates (H/J) in the manuscript, authors have not exactly mentioned the type of aggregates formed in different cases, on changing the substituent.
6. Along with temperature-dependent emission, it is ideal to show the temperature dependent lifetime profiles as well. Temperature-dependent lifetime plots are missing to confirm the nature of emission.
7. Is there any prompt component present in the lifetime profile (Figure 2h). What does it ascribed to?
8. Please recheck the legends of Figure 3b.
9. In line 168, the figure authors points to seems to be wrong; is it Figure 2g or 2h
10. There is slight contradiction between the figures Figure 4d and 4e. The lifetime decay plot suggests TpPI has longest lifetime, however, in the bar-diagram, it has very lesser lifetime compared to other derivatives. Authors please explain this.
11. With respect to the present work, which shows stable phosphorescence after mechanical grinding with afterglow as long as few seconds, author can cite the following articles: Adv. Mater. 2022, 34, 2206712; Nat. Commun. 2023, 14, 4720; Adv. Funct. Mater. 2020, 30, 2003693; Chem. Sci. 2022, 13, 10011-10019.
12. There are some minor typos: In the figure caption, 3e, 'e' has to be made bold, in line 356 'idea' has to be replaced by 'ideal'.

Reviewer #2 (Remarks to the Author):

This manuscript by Guo et al. reports the long-lived emission from the carbonyl-substituted triphenylene derivatives in the amorphous solid and crystalline state due to the formation of the molecular dimer through π - π stacking intermolecular interactions. The red long-lived emission from the amorphous materials was used to demonstrate nanoparticle-based bioimaging. The author discussed red long-lived emissions with a few 100 ms lifetimes without quantum yield information or decay rates. Hence, it is not clear what causes this red emission in triphenylene derivatives, as molecular dimer/aggregate formation does not fully explain it. The study showed that cellular imaging is possible with 488 nm excitations and red channel emission by a confocal microscope. However, the benefits of afterglow bioimaging without background information were not demonstrated. Considering the recent advancement of red afterglow emission from various amorphous solids or crystalline materials based on molecular dimers or aggregates, this manuscript lacks novelty in terms of

afterglow performance and dimer-based triplet stabilization mechanism. Therefore, this manuscript is not suitable for publication in Nature Communications in its current form.

The comments are

1. It is unclear why there is a drastic decrease in the T1 level (Fig. 1c and Table S3) after forming the π - π stacked dimer.
2. Native triphenylene has a small S0-S1 transition dipole moment due to which k_f and k_{isc} are comparable, leading to a large ISC yield ($\Phi_{isc} \sim 90\%$ at RT; J. Phys. Chem. 1969, 73, 4356) and $k_{nr}+k_q$ dominates over the k_p . Therefore, the RTP yield is very small. After substitution with the carbonyl groups, the S0-S n transition dipole may facilitate the k_p and improve the yield. Therefore, a comparative study based on Φ_{isc} values of native Tp and other TpPX is necessary to confirm the carbonyl or halogen substitution contribution for k_p .
3. From the concentration-dependent study in dioxane, why it is obvious that the emission intensity of TpPBr (Fig. 3a) initially decreases with concentration and then increases. The absorption spectra of TpPBr clearly show that J-aggregate formation is possible due to the appearance of a new peak in the higher wavelength regions. However, the aggregated absorption peak is below 400 nm in solution at a higher concentration compared to a broad onset on the absorption over 400 nm of solid powder (Fig. S15). Are these two aggregates the same or different? Determination of the nature of the aggregate in the highly concentrated solution, crystals, and amorphous powder is necessary.
4. The author discussed that the amorphous has no crystallinity, so why does the strong π - π stacking exist in the amorphous state, or is it different from the crystal state? If the nature of the aggregate is different, then why are the delayed luminescence spectra at RT almost identical (Fig. S10a).
5. Why does -OMe substitution on carbonyl phenyl not produce afterglow emissions compared to halogen substitution?
6. Show the absorption spectrum of water-dispersed nanoparticles. Can it be possible to excite at 488 nm? If the lifetime of amorphous TpPBr in the air is <2 ms over 600 nm, then how do nanoparticles of TpPBr@F127 in an aqueous medium show a longer lifetime (>200 ms)? Is the origin of long-lived emission from the nanoparticles due to microcrystal parts or completely from an amorphous state?
7. Even though nanoparticle-based bioimaging was demonstrated, no afterglow imaging was shown; it is simple cellular imaging using 488 nm excitation and red channel emission. Afterglow bioimaging is required for the benefit of autofluorescence-free images.
8. If the long-lived red emission of the crystals is mainly from the dimers of TpPBr, TpPF, and TpPMe, why does a native T1 emission band exist in the delayed spectra of crystals at RT?

Other minor points:

9. All the long-lived decay profiles show multi-exponential behaviors. Hence, it is important to analyze the different components at different emission wavelengths.
10. What is the effect of fluorescence lifetime because molecular packing via H or J-aggregation can affect the radiative decay time from singlet excited states?
11. Change Figure S6 y-axis legend to Normalized absorbance/ em. intensity
12. Significant absorption from 350-800 in Figure S6b is probably due to the negative absorption data normalized in Figure S6b.
13. Why is the peak centered at around 550 nm in Fig. S7 at 100K more intense compared to 77K?
14. The comparative Table S1 and S6 must be reconstructed with lifetime, yield, and wavelength information.
15. Correct 2nd d1 in Table S5.

Reviewer #3 (Remarks to the Author):

In this work, Yang groups developed phenyl(triphenylene-2-yl)methanone based long and persistent RTP molecules, which glows on a second time scale in crystalline, amorphous states, water and heating conditions. The work is interesting, however, these are the following points that need to be

addressed and clarified.

Major comments:

1. Authors mentioned in the introduction section (line 67-69) that "delicate intermolecular interactions and oxygen barrier properties. Accordingly, some interesting RTP properties sensitive to mechanical force, oxygen, or water have been developed based on these delicate interaction [24-27]." If there are reports where scientists have already designed RTP molecules which are sensitive to mechanical force, oxygen and water, then what is the novelty of the present work?

2. Authors showed and also discussed phosphorescence after-glow lifetime of up to second time-scale. However, it is not reflected in their decay profiles (showed in several places), where I can see the decay profile reaches flat after 20 ms.

I request authors should collect decay profiles in several 100 ms or in second time scale to confirm the after glow phenomenon in second time-scale. Lifetime decay profile is the ultimate proof whether it is glowing in the second time scale or not.

Also most of the delayed spectra reported here, where delay has been given only 8 ms. If it is glowing in the second time-scale, then the author should observe delayed spectra even in a 100 ms delayed case. Thus, I request authors should measure and report delayed emission spectra for at least one sample (TpPBr) collected in various delay time (ranging from 5 ms to 100 ms).

3. Although it is suggested that 77 K is not the good choice to measure the phosphorescence spectra, did the author try to measure in 77 K also?

Author measured the low temperature phosphorescence using dioxane as a solvent at 0 degree celcius. I guess dioxane is not transparent at 0 degree, as dioxane is not considered as glass freezing solvent. Authors should choose some glass forming solvent instead of dioxane.

4. "Under the same concentration of 10.0 mM, it was found that a new absorption peak clearly appeared at around 370 nm in the UV-visible absorption spectra of TpPBr in 1,4-dioxane solution (Fig. 3b). These results indicate that the TpPBr aggregated into dimers not excimers in a high concentration of 10.0 mM."

"Thus, these results demonstrate that the red phosphorescent emission with the maximum peak at about 610 nm in the crystal and the amorphous TpPBr samples originated from the dimers."

From the above mentioned studies in the aggregated states (high concentration in liquid, crystalline and amorphous), it is confirmed that phosphorescence emission is coming from the aggregated state but not from the monomeric state. But I did not understand why the authors think that it originated from dimeric state, not from trimer, tetramer and so on. Even in the amorphous condition, one cannot rule out higher aggregated states, as I am sure the particle dimension in the amorphous state is micron size. Also, in the aggregated particles in di-oxane there is no control over the number of particles forming aggregate even in the aggregates in dioxane (Authors can find out the dimension from DLS studies). Authors also showed strong pi-pi stacking interaction in the crystalline states, but it does not mean that dimers are only possible in the aggregated state.

5. In line no. 269-275 authors mentioned "As revealed through single-crystal XRD of four compounds, including TpP, TpPF, TpPBr, TpPI, and TpPMe, their tight dimer structures were confirmed and ascribed as H- or J- aggregates by analyzing their molecular packing structures with the shortest distance of triphenylene plane in the range of 3.390 ~ 3.554 Å (Supplementary Table 4-5 and Supplementary Figs. 12".

I can see in all the cases the arrangement is head to tail fashion, so apparently it is J-aggregate type not H aggregate. However, it is better not to mention J or H aggregates unless there is a clear signature like shift in absorption spectra etc. Surely, it cannot be H aggregate, as generally H aggregates are non-luminescent in nature.

12/4/2023

Response to the reviewers' comments and suggestions

Manuscript submitted to *Nature Communications*

Manuscript ID: NCOMMS-23-30644

Title: Visible-light-excited robust room temperature phosphorescence and red afterglow derived from a single-component dimer luminophore in the amorphous state

Authors: Danman Guo, Wen Wang, Kaimin Zhang, Jinzheng Chen, Yuyuan Wang, Tianyi Wang, Wangmeng Hou, Zhen Zhang, Huahua Huang, Zhenguo Chi, and Zhiyong Yang*

We sincerely thank the reviewers for their valuable comments. Corrections have been made according to the reviewers' suggestions (highlighted in yellow in the revised manuscript) and are explained as follows:

Response to the Comments and Suggestions of Reviewer 1

Thank the reviewer for the appreciation of our work: "*The manuscript has enough novelty and suitable to publish in this journal. However, the author should address the following points.*"

1. Since the system exhibits quite efficient afterglow, authors can think about adding the term 'Afterglow' in the title to elevate the impact on readers.

Response: Thanks for the reviewer's valuable suggestion. The title has been changed to "Visible-light-excited robust room temperature phosphorescence and red afterglow derived from a single-component dimer luminophore in the amorphous state" in the revised manuscript.

2. The writings on schematics of Figure 1b are not visible, please change them.

Response: Thank the reviewer for pointing out this. We have improved it as below.

Revised Fig. 1: Schematic representation for visible-light-excited robust RTP based on dimer luminophore. ... **b**, Molecular design and the dimer formation for persistent RTP. The red plane, such as the triphenylene core, possesses an electronic configuration of $^3(\pi, \pi^*)$ in an excited state and subsequent slow intersystem crossing (ISC). The blue unit, such as phenyl carbonyl, has an electronic configuration of $^3(n, \pi^*)$ with a fast ISC rate. ...

3. What does the feature from 500-560 nm in Figure S7 signifies, monomeric fluorescence or phosphorescence. And, why does the changes for this band is irregular in temperature dependent emission spectra.

Response: Thanks a lot for this question, which encourages us to obtain a deep understanding of these new amorphous RTP systems. Firstly, the delayed emission of powder at 500-560 nm (516 and 560 nm) only appeared at the low temperature of 77 K (Figure S8 and R1a), which excluded its possibility of thermal activated delayed fluorescence (TADF). Moreover, as shown in the responsive Figure R1a, these delayed emission peaks at 516 and 560 nm were significantly different from either the prompt or delayed peaks of the isolated molecule (10^{-4} M, light-blue line) and the aggregates (10^{-2} M, blue line). This result excluded

the possibility of TADF/triplet-triplet annihilation (TTA) and monomer phosphorescence (the first triple excited state (T_1), peaks at 480 and 520 nm) of these peaks. Therefore, the peaks at 516 and 560 nm may be the emission of aggregates, such as that from the high-lying triplet states of aggregates (T_n^{dimer} , generally T_2^{dimer}).

In order to understand these peaks clearly, we tested the delayed luminescence spectra of TpPBr in the crystalline and amorphous states in different atmospheres (in air/in a vacuum) at low temperatures. The results showed delayed peaks at 516 and 560 nm only appeared at low temperatures and in a vacuum, as shown in the responsive Figure R1a. These peaks could be observed both at 100 K and 77 K in a vacuum for the crystalline powder. However, it is noted that the amorphous powder mainly showed the emission peaks of T_1 (480 and 520 nm) at 77 K in a vacuum, while T_2^{dimer} peaks (516 and 560 nm) at 100 K (Figure R1b). Compared to the T_1 state of aggregates (T_1^{dimer}), the T_2^{dimer} has a larger energy gap for the transition back to the ground state (S_0). It is more significantly affected by non-radiative transitions and oxygen quenching. Therefore, significantly delayed emission from T_2^{dimer} can only be seen at low temperatures in a vacuum. When the temperature increased from 77 K to 100 K, the phosphorescence of T_2^{dimer} enhanced in crystalline powder or appeared in amorphous powder due to the thermal equilibrium of excitons converting from T_1^{dimer} to T_2^{dimer} states. As the temperature further increased, the non-radiative transitions of T_2^{dimer} excitation were promoted, and its emission was weakened (reference such as *Mol. Phys.* **27**, 969-979 (1974); *J. Phys. Chem.* **91**, 819 (1987); *Chem. Rev.* **112**, 4541-4568 (2012)). Thus, the change for the band at 500-560 nm is irregular in the temperature-dependent emission spectra of Figure S8.

At low temperatures and in a vacuum, some systems could observe a blue-shifted emission directly from a high-lying excited state due to the efficient suppression of internal conversions and non-radiative transitions. It has been reported in the literature that compounds containing halogen or carbonyl groups exhibited double phosphorescence at 77 K due to the large energy gap between excited states (such as 0.19 eV in *Nat. Commun.* **8**, 416 (2017) and 0.42 eV in *Angew. Chem. Int. Ed.* **61**, e202205556 (2022)). The energy gap between 515 nm (T_2^{dimer}) and 600 nm (T_1^{dimer}) was measured to be 0.34 eV, making it possible to observe the emission from the T_2^{dimer} state. To verify this hypothesis further, spin-orbit coupling matrix elements were calculated at the time-dependent density functional theory (TDDFT) by Orca based on the M062X TZVP basis set, as shown in Figure R1c. The results showed that the energy gaps of T_2 - T_3 and T_4 - T_5 were 0.21 eV and 0.34 eV, respectively, which were two of the largest among the energy gaps between T_n s. The

total spin-orbit coupling (SOC) matrix elements of S_0 and T_3 - T_{10} can be up to 55.5 cm^{-1} , indicating that the radiative transition from T_n to S_0 was possible.

The time-resolved spectral decay of TpPBr crystalline powder in different conditions of atmospheres and temperatures further confirmed this mechanism, as shown in responsive Figure R1d-i. The phosphorescent peak of T_1 was seen at the delayed time of $\sim 50 \text{ ns}$, while the T_1^{dimer} peak could be observed a bit later at $\sim 80 \text{ ns}$ when TpPBr crystals were measured in the air at 298 K (Figure R1d and g, added into the revised Fig. 3g-h in the revised manuscript). If detected in the vacuum at 298 K , the T_1^{dimer} peak would appear earlier at 60 ns (Figure R1e). However, no peaks for T_2^{dimer} (typical peak at 560 nm) could be seen in these conditions. The significant emission peaks (560 nm) for T_2^{dimer} can only be observed after 50 ns at 77 K in a vacuum, which appeared simultaneously and decayed along with the T_1^{dimer} peak at $\sim 600 \text{ nm}$ (Figure R1f). These results indicated that these peaks (T_2^{dimer}) were associated with the T_1^{dimer} peak, which further confirmed it as the emission from the high-lying excited state of aggregates.

Responsive Figure R1: a) Normalized prompt luminescence spectra of TpPBr in 1,4-dioxane and its crystalline powder measured in air at 298 K (upper, sol.: $\lambda_{\text{ex}} = 310$ nm; crys.: $\lambda_{\text{ex}} = 365$ nm); Delayed luminescence spectra of TpPBr in 1,4-dioxane in air and crystalline powder in a vacuum/in air at 298/77 K (lower, $\lambda_{\text{ex}} = 365$ nm, delayed 8 ms). b) Normalized delayed luminescence spectra of TpPBr in crystalline and amorphous state in a vacuum or air ($\lambda_{\text{ex}} = 365$ nm, delayed 8 ms). c) Spin-orbit coupling calculations of TpPBr-dimer based on the single crystal conformation. d-i) Normalized time-resolved emission

spectroscopy (TRES) of TpPBr crystalline powder in air/in a vacuum at 298/77 K in nanosecond scale ($\lambda_{\text{ex}} = 405 \text{ nm}$). (d, g) Air, 298 K; (e, h) vac., 298; (f, i) vac., 77 K.

4. Authors should also record lifetime of the band before the phosphorescence band from the dimer to understand its origin unambiguously.

Response: Thanks a lot for the valuable suggestion. The lifetime of TpPBr in 1,4-dioxane solutions with incremental concentrations has been measured to analyze the dimer formation process, as shown in the added Figure S9 below. The fluorescence lifetime (τ_f) is given by: $\tau_f = \frac{1}{k_f + k_{\text{nr}} + k_{\text{isc}}}$. Where k_f , k_{nr} , and k_{isc} are the rate constants of fluorescence, non-radiative transitions, and intersystem crossing, respectively. As shown in Fig. 3a-b, dimers were partially formed as the solution concentration gradually increased from 0.5 to 1 mM. Thus, fluorescence lifetime increased due to the general longer lifetime of dimers, as shown in added Figure S9. When the concentration further increased ($>1 \text{ mM}$), dimer aggregates formed, and the intersystem crossing was promoted, leading to dimer phosphorescence. The fluorescence lifetime then decreased due to the k_{isc} increase, which became close to that of the crystalline powder of TpPBr (added Figure S8). The discussion about this was also added to the revised manuscript: “The relative fluorescence lifetime at about 460 nm increased first and then decreased along with the concentration change. It also revealed the dimers/excimers formed and subsequently aggregated in the solution (Supplementary Fig. 9).” in Lines 184-186 in the revised manuscript.

Added Figure S9. Fluorescence intensity decay profiles of TpPBr in 1,4-dioxane solutions with incremental concentrations or its crystalline powder (298 K, air, $\lambda_{\text{ex}} = 405 \text{ nm}$).

5. Although, there is a sustenance on the type of aggregates (H/J) in the manuscript, authors have not exactly mentioned the type of aggregates formed in different cases, on changing the substituent.

Response: Thank the reviewer for this valuable question, which encourages us to obtain a deep understanding of the aggregates. Compared to the others, the fluorescence signals of TpPI and TpPOMe were much weaker. It is widely recognized that fluorescence will decrease in H-aggregates. Therefore, we initially deduced that the former could be composed of J-aggregates, while the latter may contain H-aggregates. The stacking modes of their single crystals were carefully analyzed using the molecular exciton theory, and the results were summarized in the updated Table S7 in the revised SI. As shown in Fig. 3d and S12, all TpPX compounds adapted a parallel stacking mode and formed dimers in their crystals. Among these compounds, TpPOMe and TpPI adopted a head-to-head parallel stacking mode, while other compounds (TpP, TpPF, TpPBr, and TpPMe) took a head-to-tail stacking mode.

In order to identify their exact aggregation type, the angles (θ) between the transition moment of the molecule and the interconnection of the molecular centers were calculated based on their single-crystal structures, as summarized in Table S7. The resulting angles for the dimers of TpP and TpPF were revealed to be 43.8° and 16.4° , respectively, which are lower than the critical angle (54.7°) for H-aggregation. So, the aggregates are purely J-aggregate for TpP and TpPF, which possessed good fluorescence quantum yields (QY) (Table S3). In contrast, the two types of dimer in the crystals of TpPI and TpPOMe both belong to H-aggregate due to all the angles being above 54.7° , leading to weak fluorescence with low QY (Table S3). Regarding TpPBr and TpPMe, they also contain two types of dimers but with different angles above and below 54.7° simultaneously. They thus formed mixed H/J-aggregates in their single crystals, resulting in moderate fluorescence QY (Table S3). These results correspond to the experimental fluorescence properties of these TpPX compounds. For phosphorescence, the relationship became irregular partially due to the heavy-atom effect of bromo- and iodo- substituents.

The relative sentences, including “Correspondingly, two types of dimers of TpPBr could be formed between the molecular packing structures, which were analyzed to be H- and J- aggregation using the molecular exciton theory[49], respectively (Supplementary Fig. 11-12 and Table 7).” and “As revealed through single-crystal XRD of six TpPX compounds and Tp, their tight dimer structures were confirmed and ascribed as H- or J- aggregates by analyzing their molecular packing structures with the distance of triphenylene plane in the range of $3.374 \sim 3.695 \text{ \AA}$ ” have been modified in Lines 233-236 and 282-286 in the revised manuscript. The relative discussion has also been added to the revised manuscript as: “The Tp core itself

showed an H-aggregation in the crystal. After being substituted by the phenyl carbonyl unit, only TpPI and TpPOME adopted H-aggregation in their two dimer types in the crystals. The dimers of TpP and TpPF belonged to J-aggregates. Regarding TpPBr and TpPMe, they contained two types of dimers and formed mixed H/J-aggregates in their crystals.” in Lines 287-291 in the revised manuscript.

Figure S11. Schematic representation of the aggregation models. The blue sphere and purple plane are the centroid and plane of the triphenylene core, respectively. And d_1 represents the vertical distance between two adjacent triphenylene planes, while d_2 is the centroid distance of two adjacent triphenylene planes. The red arrow describes the transition dipole moment of the monomer in the dimer. The green line is across the centroid of the monomer in the dimer. And the angle between the transition dipoles and the interconnected axis is indicated by θ . The transition dipole moment is calculated using the Gaussian 09W package.

Fig. 3: Characterization of the formation of TpPBr dimer for its robust RTP. ... d, The packing structure of TpPBr dimer in its crystal structure. ...

Figure S12. Crystal structures of TpPX. Distance represents the vertical distance between two adjacent triphenylene planes.

Added Table S5. The aggregation models of TpPX. Results are calculated based on their single crystal structures. Here, d_1 represents the vertical distance between two adjacent triphenylene planes, while d_2 is the centroid distance of two adjacent triphenylene planes. The angle between the transition dipoles and the interconnected axis is indicated by θ .

Compound-dimer	$d_1 / \text{\AA}$	$d_2 / \text{\AA}$	θ	transition electric dipole moments of S_0 - S_1 (Au)	H/J-aggregation
TpP	3.390	3.613	43.8	(0.1271, -0.0780, -0.0935)	J
TpPF	3.427	3.631	16.4	(-0.1434, -0.0841, 0.0960)	J
TpPBr	3.695	4.496	39.0	(-0.0498, -0.0752, -0.0994)	J
	3.554	3.823	76.9	(-0.0498, -0.0752, -0.0994)	H
TpPMe	3.590	4.823	34.3	(0.0414, 0.0642, 0.0954)	J
	3.472	3.748	68.7	(0.0414, 0.0642, 0.0954)	H
TpPI	3.399	4.483	66.4	(-0.0153, 0.0761, 0.1739)	H
	3.481	4.483	71.8	(0.0829, -0.0286, -0.0261)	H
TpPOMe	3.380	4.009	64.0	(-0.0829, 0.1141, -0.1253)	H
	3.480	4.564	66.0	(0.0736, -0.1177, -0.1160)	H
Tp	3.374	5.274	65.1	(0.0025, 0.0052, -0.0014)	H

6. Along with temperature-dependent emission, it is ideal to show the temperature dependent lifetime profiles as well. Temperature-dependent lifetime plots are missing to confirm the nature of emission.

Response: Thanks for this valuable suggestion. The PL decay profiles of TpPBr powder were measured and added as Figure S8b, and the lifetimes were summarized in Table S5 in the revised SI. When the temperature decreased from 300 K to 77 K, the lifetime increased with the exception of a slight drop from 100 to 77 K. A meticulous analysis of delayed luminescence spectra and time-resolved emission spectroscopy of TpPBr samples in various conditions has been performed and detailed discussed above in Response 3. It is revealed that the delayed emission of T_1 , T_2^{dimer} , and T_1^{dimer} could be observed in TpPBr crystalline powder at 100 and 77 K, which are located at 520, 560, and 600 nm, respectively. Therefore, the lifetime at 600 nm became a bit decrease as the temperature dropped from 100 to 77 K due to its complex consistent of delayed emission of T_2^{dimer} and T_1^{dimer} . The relative sentence has been changed to “The phosphorescence characteristic of TpPBr was also supported by the continuous increase in its emission intensity and lifetime with temperature decreasing (Supplementary Fig. 8 and Table 5).” in Lines 134-136 in the revised manuscript.

Revised Figure S8. a) Delayed luminescence spectra of TpPBr measured at temperatures from 77 to 300 K in a vacuum ($\lambda_{\text{ex}} = 365$ nm). b) PL intensity decay profiles of crystalline TpPBr powder at a peak of 605 nm in a vacuum ($\lambda_{\text{ex}} = 340$ nm).

Added Table S5. Lifetimes of TpPBr powder at the 605 nm peak at temperatures from 77 to 300 K in a vacuum ($\lambda_{\text{ex}} = 340$ nm).

Temperature / K	τ / s @ $\lambda_{\text{em}} = 605$ nm (Å)		
77	1.72 (45%)	0.55 (30%)	0.20 (25%)
100	1.69 (50%)	0.61 (39%)	0.16 (11%)
150	1.69 (52%)	0.50 (37%)	0.13 (11%)
200	1.63 (53%)	0.51 (36%)	0.12 (11%)
250	1.29 (56%)	0.47 (33%)	0.13 (11%)
300	0.12 (65%)	0.42 (35%)	-

7. Is there any prompt component present in the lifetime profile (Figure 2h). What does it ascribed to?

Response: Thanks for this valuable comment. As shown in Fig. 2h, the lifetimes showed not a direct decay from the maximum intensities but a steep decline to a certain level and then another slow decay. This phenomenon was attributed to the short-lived components in the lifetime decay. From the time-resolved emission spectroscopy (added Figures 3g-h and S10), fluorescence was the main component of the beginning of the decay. At the same time, the long-lived phosphorescence eventually became the main

component after about 10 ns. Therefore, short-lived components were present at the lifetime profiles of phosphorescent peaks, mainly ascribed to be emission from the first singlet state of aggregates (S_1^{dimer}).

Added Figure S10. Normalized time-resolved emission spectroscopy (TRES) of TpPBr in a crystalline or amorphous state in air/in a vacuum at 298/77 K at nanosecond scale ($\lambda_{\text{ex}} = 405 \text{ nm}$). (a, c) Crys., vac., 298 K; (b, d) crys., vac., 77 K; (e, g) amor., air, 298 K; (f, h) amor., vac., 298 K.

8. Please recheck the legends of Figure 3b.

Response: Thank the reviewer for the careful reading. We have corrected it as follows:

Revised Fig. 3b. UV-vis absorption of TpPBr in 1,4-dioxane solutions with incremental concentrations. Local magnification of the absorption in the 350-500 nm range was inset for comparison.

9. In line 168, the figure authors points to seems to be wrong; is it Figure 2g or 2h?

Response: Thanks for pointing out this mistake. We have corrected it to “Fig. 2h”.

10. There is slight contradiction between the figures Figure 4d and 4e. The lifetime decay plot suggests TpPI has longest lifetime, however, in the bar-diagram, it has very lesser lifetime compared to other derivatives. Authors please explain this.

Response: Thanks for the comment. Actually, Fig. 4d showed the phosphorescent lifetime of TpPI in the amorphous state *in the air*. In contrast, Fig. 4e represents the lifetime of TpPI in the amorphous state *in a vacuum* and the crystalline state *in air*, respectively. To avoid misunderstanding, the legend of Fig. 4e has been improved as “e, Comparison between the lifetime of TpPX powder in the crystalline (in the air) and the amorphous (in the vacuum) states ($\lambda_{\text{ex}} = 415 \text{ nm}$)” in the revised manuscript.

Interestingly, amorphous TpPI powder exhibited almost the same phosphorescence lifetime in air or a vacuum, as shown in Fig. 4d-e. It means that amorphous TpPI powder shows good resistance to oxygen in the air. We speculate that the dimer stacking is stronger for TpPI molecules than the other TpPX derivatives due to its strong intermolecular interactions after the substitution of heavy-atom Iodine. Therefore, most TpPI molecules formed partial dimer stacking in the amorphous state, which could be clearly demonstrated by almost no delayed emission in the 470-570 nm range for the isolated molecules (Figure S16-17). The phosphorescent lifetime of amorphous TpPI powder in air is thus much longer than that of other powders, as shown in Fig. 4d.

11. With respect to the present work, which shows stable phosphorescence after mechanical grinding with afterglow as long as few seconds, author can cite the following articles: Adv. Mater. 2022, 34, 2206712; Nat. Commun. 2023, 14, 4720; Adv. Funct. Mater. 2020, 30, 2003693; Chem. Sci. 2022, 13, 10011–10019.

Response: Thanks for the reviewer's valuable suggestion. These present works have been added and cited in the revised manuscript as follows: *Adv. Mater.* 2022, 34, 2206712 (reference 19); *Nat. Commun.* 2023, 14, 4720 (reference 20); *Adv. Funct. Mater.* 2020, 30, 2003693 (reference 21); *Chem. Sci.* 2022, 13, 10011-10019 (reference 22).

12. There are some minor typos: In the figure caption, 3e, 'e' has to be made bold, in line 356 'idea' has to be replace by 'ideal'.

Response: Thank the reviewer for his/her careful reading. We have corrected these mistakes in the revised manuscript.

Response to the Comments and Suggestions of Reviewer 2

Considering the recent advancement of red afterglow emission from various amorphous solids or crystalline materials based on molecular dimers or aggregates, this manuscript lacks novelty in terms of afterglow performance and dimer-based triplet stabilization mechanism. Therefore, this manuscript is not suitable for publication in Nature Communications in its current form.

Response: It may be a misunderstanding due to our poor expression here. There is great progress in organic materials with room temperature phosphorescence (RTP) and afterglow. As stated in the introduction of

our manuscript, a robust persistent RTP luminophore system whose phosphorescent emission is not restricted to the crystalline state or other rigid environments with cautious treatment is highly demanded for its practical application. A few organic amorphous systems (references 16-22) with robust RTP and afterglow are reported through host-guest complex or doping method. However, the potential risk of inevitable phase separation in these systems is an avoidable problem that restricts their application. Additionally, the excitation wavelength for these reported systems is in the ultraviolet region, which is unsuitable for bioimaging applications. Here, we report the first example of a single-component system (TpPX) with robust persistent RTP emission and afterglow in various aggregated forms, such as crystal, fine powder, and even amorphous states. Our experimental data reveal that the vigorous RTP emissions rely on their tight dimers based on strong and large-overlap π - π interactions between polycyclic aromatic hydrocarbon (PAH) groups. The dimer structure can offer not only excitons in low energy levels for visible-light excited red long-lived RTP but also suppression of the nonradiative decays even in an amorphous state for good resistance of RTP to heat (up to 70 °C) or water. Furthermore, we demonstrate the water-dispersible TpPBr nanoparticle with persistent red RTP over 600 nm and a lifetime of 0.22 s for visible-light excited cellular and in-vivo imaging, prepared through the common microemulsion approach without overcaution for nanocrystal formation. As summarized in Table S8 in the revised SI, such a result is the rare organic deep red (over 600 nm) RTP system with a lifetime of over 0.1 s for in-vivo bioimaging excited by visible light. Therefore, the findings open the opportunity for developing single-component luminescent materials with robust phosphorescence in an amorphous state through an effective strategy of forming a tight dimer.

1. It is unclear why there is a drastic decrease in the T1 level (Fig. 1c and Table S3) after forming the π - π stacked dimer.

Response: There should be a misunderstanding here due to our unclear description of these T_1 levels. Actually, there is a new triplet level (T_1^*) after forming the π - π stacked dimer in the TpPX powder, which is different from the original triplet level (T_1) of the TpPX monomer. Therefore, the symbols of the newly formed triplet level in Table S4 have been improved to T_1^* , making it much clearer. Benefiting from the large planar conjugated structure of triphenylene, TpPX tends to form dimers via strong π - π stacking. Intermolecular electronic coupling between the TpPX molecules in the dimer could modulate the energy level distribution, thereby generating a new low energy level of T_1^* for the dimer. Therefore, triplet excitons

can be stabilized in the low energy level T_1^* state, resulting in a significant redshift of phosphorescence emission compared to the isolated molecule. This phenomenon is similar to the new low energy level of S_1^* of the dimer in fluorescence emission, which also generates a red-shifted fluorescence. Actually, the emission from the T_1 state of TpPX still could be observed in these compounds, which are relatively weak and located at the range of 470-570 nm (Fig. 2f). The pronounced redshift of phosphorescence in organic materials after aggregation has been thoroughly investigated in recent years (references such as *Nat. Commun.* **9**, 840 (2018); *Angew. Chem. Int. Ed.* **57**, 7997 (2018); *Chem. Sci.* **10**, 179-184 (2019); *J. Am. Chem. Soc.* **142**, 1153-115 (2020); *Nat. Commun.* **11**, 4802 (2020); *Nat. Commun.* **13**, 2658 (2022); *Angew. Chem. Int. Ed.* **62**, e202214908 (2023)).

2. Native triphenylene has a small S_0 - S_1 transition dipole moment due to which k_f and k_{isc} are comparable, leading to a large ISC yield ($\Phi_{isc} \sim 90\%$ at RT; *J. Phys. Chem.* 1969, 73, 4356) and $k_{nr}+k_q$ dominates over the k_p . Therefore, the RTP yield is very small. After substitution with the carbonyl groups, the S_0 - S_n transition dipole may facilitate the k_p and improve the yield. Therefore, a comparative study based on Φ_{isc} values of native Tp and other TpPX is necessary to confirm the carbonyl or halogen substitution contribution for k_p .

Response: Thanks a lot for such a valuable suggestion. According to the reviewer's advice, the photophysical parameters of Tp (Figure R2) and TpPX in air, including lifetime, quantum yield, and different rate constants, have been carefully studied and summarized in the added Table S3. It is evident that the quantum yield of intersystem crossing (Φ_{isc}) of TpPX is greater than that of Tp, indicating that the incorporation of carbonyl can significantly enhance their ISC efficiencies. Moreover, the phosphorescence quantum yield (Φ_p) has a close correlation with the phosphorescence rate constant (k_p), as shown in the below equation (1). In accordance with the equations (2-3) below, a more significant transition dipole moment, a larger spin-orbit coupling (SOC) matrix element, and a smaller energy gap (ΔE_{ST}) can promote k_p , resulting in a higher Φ_p .

Φ_p and k_p are approximately expressed as (*Ber. Bunsengesellschaft Phys. Chem.* **67**, 46 (1963); *Phys. Chem. Chem. Phys.* **16**, 14523 (2014); *J. Phys. Chem. Lett.* **9**, 4251 (2018)):

$$\Phi_p = \Phi_{isc} \times \frac{k_p}{k_p + k_{nr} + k_q} = \Phi_{isc} k_p \tau_p \quad (1)$$

$$k_p \propto \lambda_p^2 (\sum_m \mu_{s_m - s_0} \lambda_m)^2 \quad (2)$$

$$\lambda_m \approx |\langle \psi_m^1 | \overline{H_{so}} | \psi_1^3 \rangle| / E_{S_m - T_1} \quad (m \geq 2) \quad (3)$$

Where λ_p is the phosphorescence energy, $|\langle \psi_m^1 | \overline{H_{so}} | \psi_1^3 \rangle|^2$ is the spin-orbit coupling between S_m and T_1 , and $E_{S_m - T_1}$ is the energy gap between S_m and T_1 , $\mu_{S_m - S_0}$ is transition dipole moment between S_m and S_0 .

The introduction of carbonyl results in slight charge transfer characteristics and promotes the transition dipole moment in TpPX compounds. And the SOC of TpPX was calculated using Gaussian+orca based on TDDFT (M062X TZVP). Analysis of the calculated results indicated a larger SOC of TpPX for promoting phosphorescence (Figure S15). Therefore, the k_p of Tp is only 0.029 s^{-1} , whereas that of TpPX is increased to $0.106\text{-}0.756 \text{ s}^{-1}$, as shown in Table S3. As it possesses different aggregation modes in the crystal, the halogen substitution on TpPX compounds has an irregular effect on the k_p . For the same reason, all TpPX compounds exhibit a higher Φ_p than Tp, except for TpPI and TpPOMe. Due to the promotion of k_p , the phosphorescence lifetimes of TpPX became a bit shorter than Tp. Overall, introducing the carbonyl group increases the proportion of phosphorescence emission from the long-wavelength aggregation state and its quantum yield (Table S3). The relative discussion “The calculated data also indicated that hole/electron located on both molecules in the dimer structure and a larger SOC of TpPX for promoting phosphorescence (Supplementary Fig. 14-15). The experimental quantum yield of ISC (Φ_{isc}) and phosphoresce rate constant (k_p) of TpPX were increased when compared to the Tp core itself (Supplementary Table 3).” and “These results illustrate the crucial roles of the triphenylene unit in forming the stable dimers and the carbonyl group in promoting ISC to achieve such robust and highly desirable RTP properties in different states of TpPBr.” has been modified in Lines 249-256 in the revised manuscript.

Added Table S3. Photophysical parameters of Tp and TpPX powder at room temperature in air.

Chromophore	Φ_f (%) ^a	Φ_{isc} (%) ^b	λ_f (nm)	τ_f (ns) ^a	k_f (10^7 s^{-1}) ^c	k_{isc} (10^7 s^{-1}) ^d	Φ_p (%) ^e	λ_p (nm)	τ_p (ms) ^f	k_p (s^{-1}) ^g
Tp	34.96	65.04	425	9.54	3.664	6.818	2.18	590	1147	0.029
TpP	14.40	85.60	480	12.29	1.171	6.963	10.25	610	390	0.307
TpPF	15.28	84.72	463	6.33	2.413	13.378	4.25	603	470	0.107
TpPBr	6.24	93.76	537	5.60	1.115	16.742	6.71	610	190	0.376
TpPI	0.62	99.38	500	2.46	0.252	40.328	0.76	608	72	0.106

TpPMe	14.82	85.18	530	6.25	2.369	13.621	9.26	600	390	0.279
TpPOMe	1.80	98.20	500	8.59	0.209	11.435	1.93	630	26	0.756

^a $\lambda_{\text{ex}} = 405 \text{ nm}$;

^b Considering $k_f \gg k_{\text{ic}} (\text{RT})$, $k_{\text{isc}} (\text{RT}) \gg k_{\text{ic}} (\text{RT})$, it can be approximated as $\Phi_{\text{isc}} = 1 - \Phi_f$;

^c Calculated $k_f = \Phi_f / \tau_f$;

^d Calculated $k_{\text{isc}} = \Phi_{\text{isc}} / \tau_f$;

^e Calculated from aggregates by peak fitting;

^f $\lambda_{\text{ex}} = 415 \text{ nm}$;

^g Calculated $k_p = \Phi_p \times \Phi_{\text{isc}}^{-1} \times \tau_p^{-1}$.

Added Figure S15. Theoretical calculated spin-orbit coupling (SOC) matrix element of Tp and TpPX from their single crystals.

Responsive Figure R2. Photophysical parameters of Tp in air. Normalized prompt luminescence spectra of Tp in 2-methyltetrahydrofuran (purple dashed line) and its powder (light-blue dashed line) at 298 K (sol., 5×10^{-5} M, $\lambda_{\text{ex}} = 300$ nm; crys., $\lambda_{\text{ex}} = 310$ nm). Normalized delayed luminescence spectra of Tp powder at 298 K (blue solid line) or 77 K (brown solid line) ($\lambda_{\text{ex}} = 365$ nm, delayed 8 ms). Sol., solution; Crys., crystalline.

3. From the concentration-dependent study in dioxane, why it is obvious that the emission intensity of TpPBr (Fig. 3a) initially decreases with concentration and then increases. The absorption spectra of TpPBr clearly show that J-aggregate formation is possible due to the appearance of a new peak in the higher wavelength regions. However, the aggregated absorption peak is below 400 nm in solution at a higher concentration compared to a broad onset on the absorption over 400 nm of solid powder (Fig. S15). Are these two aggregates the same or different? Determination of the nature of the aggregate in the highly concentrated solution, crystals, and amorphous powder is necessary.

Response: For the changes on the emission intensity, a Förster resonance energy transfer (FRET) from the monomer to the dimer played an important role. The TpPBr molecules are predominantly isolated in the solution of 0.01 mM, whereas the molecules predominantly formed dimers in a solution with a high concentration of 10 mM. Figure R3 shows the good overlap between the fluorescence emission of the monomer (0.01 mM) and the absorption/excitation spectrum of the dimer (10 mM), indicating the possibility of efficient FRET from the monomer to the dimer. So, as the concentration increased from 0.1 mM to 0.5 mM in Fig. 3a, a substantial number of dimers formed in the solution. Subsequently, the monomer fluorescence decreased due to the effective FRET process from the monomer to the dimer. After

that, fluorescence was enhanced as the concentration continuously increased due to the growth of the number of emissive centers (dimers).

Responsive Figure R3. Absorption spectrum, prompt luminescence spectrum, and luminescence excitation spectra of TpPBr in 1,4-dioxane solution at 298 K ($\lambda_{\text{ex}} = 310$ nm). Abs., absorbance; Exc., excitation.

Due to the detection limit of the UV-vis absorption instrument, the high-concentration solution in the UV-vis spectra of a high-concentration solution was obtained by subtracting the background of a slightly lower-concentration solution in the previous manuscript. Therefore, new fake absorption peaks at lower than 400 nm appeared in the high-concentration solution, which could not be used to identify the aggregation types. In order to obtain the accurate absorption spectra for a high concentration solution, a very thin cuvette was used. Thus, new UV-vis absorption spectra of the TpPBr solution were collected and shown in the revised Fig. 3b below using pure solvents as background. As the chromophore's concentration increased in the solution, more and more dimers formed, and the absorption onset gradually extended to 400 nm due to the enhanced intermolecular interactions between TpPBr molecules. It is noted that a new small but obvious peak at around 440 nm appeared in the high concentration ($>10^{-3}$ M) solution of TpPBr, which shows similarity to the absorption characteristics of its powder.

Revised Fig. 3b. UV-vis absorption of TpPBr in 1,4-dioxane solutions with incremental concentrations. Local magnification of the absorption in the 350-500 nm range was inset for comparison.

Additionally, we carefully compared the photophysical properties of the aggregates in different conditions, as shown in responsive Figure R4. The delayed luminescence spectra of crystalline/amorphous powders and the highly concentrated solution contained similar peaks at the same wavelengths, including phosphorescence from monomers (450 - 550 nm) and aggregates (550 - 650 nm). This indicates that aggregates were essentially alike despite the different ratios of the peaks' intensities. The highly concentrated solutions and amorphous powder encompassed a significant amount of disordered molecules. As a result, the proportion of monomers' phosphorescence was comparatively higher than that of crystalline powder (Figure R4a). Moreover, Figures R4b-c demonstrated the comparable luminescent spectra of the crystalline and amorphous powders, including delayed and prompt ones, further confirming the same aggregates in these different conditions. As discussed in Response 3 to Reviewer 1, the emission of T_1 is predominant in the range of 470-570 nm in the spectrum of the amorphous powder in a vacuum at 77 K. While the luminescence from T_2^{dimer} is predominant in crystalline powder, which facilitated by the rigidification of the molecules in the crystal. Their delayed spectra thus showed a bit of difference in a vacuum at 77 K (Figure R4b). Their prompt spectra are also similar, except for the stronger phosphorescence peak of the crystal (Figure R4c). Therefore, these results demonstrate the same aggregates formed for TpPBr in different conditions, including highly concentrated solutions, crystals, and amorphous powders.

Responsive Figure R4. a) Normalized delayed luminescence spectra of TpPBr in a crystalline and amorphous state or its solution in 1,4-dioxane (10^{-2} M) in the air ($\lambda_{\text{ex}} = 365$ nm, delayed 8 ms). b) Normalized delayed luminescence spectra of TpPBr in a crystalline and amorphous state in air or in a vacuum ($\lambda_{\text{ex}} = 365$ nm, delayed 8 ms). c) Normalized prompt luminescence spectra of TpPBr in crystalline and amorphous state in air ($\lambda_{\text{ex}} = 310$ nm).

4. The author discussed that the amorphous has no crystallinity, so why does the strong π - π stacking exist in the amorphous state, or is it different from the crystal state? If the nature of the aggregate is different, then why are the delayed luminescence spectra at RT almost identical (Fig. S10a).

Response: Based on careful analysis of the luminescence properties, especially the delayed luminescence of these TpPX in crystalline and amorphous states, we confirmed that the aggregates were the same in different conditions, including highly concentrated solution, crystals, and amorphous powders (detailed discussion could be found in Response 3 above).

Regarding the existence of strong π - π stacking in TpPX amorphous powders, the large planar conjugation structure of the Tp core is the key. Other π - π stacking in amorphous systems has been also investigated recently (such as *Nat. Commun.* **12**, 2297 (2021); *Chem. Eng. J.* **446**, 136935 (2022)). As shown in Figure S17, the delayed spectral profiles of the π - π stacking aggregates in the 580-800 nm range are almost identical for these TpPX with different substituents in their crystals. This means that the interaction in the π - π stacking of the Tp core is strong, as it is little affected by the different stacking models (H/J aggregation) in these TpPX crystals. Additionally, their strong interactions are also observed in the results of the theoretical calculation. As shown in the isosurface map of interaction region indicator (IRI) = 1.1 in Fig. 3e

and S13, a very broad green isosurface with $\text{sign}(\lambda^2)\rho$ value smaller than -0.005 was found in the middle of two Tp units in the dimer structure, revealing the remarkable van der Waals (vdW) attractive interaction between the two TpPX molecules. In addition, there are several orange spots on this green isosurface, which indicates the presence of weak repulsion between the two triphenylene planes due to the crowded π orbitals. Therefore, the formation of partial π - π dimers in the amorphous state is possible, leading to the identical delayed luminescence spectra at room temperature of amorphous and crystalline TpPX.

5. Why does -OMe substitution on carbonyl phenyl not produce afterglow emissions compared to halogen substitution?

Response: Thanks for the valuable comment. The weak RTP and afterglow emission of TpPOMe with -OMe substitution may be because of its close π - π stacking in crystal and the effective quenching processes. In the single crystals, both TpPI and TpPOMe adopted a head-to-head stacking and belonged to purely H-aggregation that will diminish luminescence. However, the stacking of TpPOMe is much closer than TpPI, as its vertical distance (d_1) and centroid distance (d_2) of two adjacent Tp planes (the closer one among its two types of dimers) are much smaller than TpPI, as shown in the added Table S7. Too compact intermolecular interaction generally quenched the emission. Moreover, taking TpPBr as a reference, the quenching of TpPOMe by oxygen and nonradiative processes was much more severe, as shown in Figure R5. At room temperature, when the atmospheric conditions changed from air to vacuum, the phosphorescence lifetimes of TpPBr and TpPOMe increased 1.11 and 1.34 times, respectively. When the temperature decreased from 298 to 77 K, the phosphorescence lifetimes of TpPBr and TpPOMe further increased by a factor of 6.96 and 9.77, respectively. These results indicated a larger oxygen quenching rate (k_q) and non-radiative rate (k_{nr}) for TpPOMe than those of TpPBr. For TpPI, the heavy-atom iodine can efficiently promote the ISC process for phosphorescence through intermolecular interactions, which overcame the k_q and k_{nr} , resulting in better phosphorescence of TpPI at room temperature.

Responsive Figure R5. Lifetimes of TpPBr and TpPOMe in crystalline state at the 600 or 572 nm peak, respectively (298 K & 77 K, in the air & a vacuum, $\lambda_{\text{ex}} = 415$ nm).

6. Show the absorption spectrum of water-dispersed nanoparticles. Can it be possible to excite at 488 nm? If the lifetime of amorphous TpPBr in the air is <2 ms over 600 nm, then how do nanoparticles of TpPBr@F127 in an aqueous medium show a longer lifetime (>200 ms)? Is the origin of long-lived emission from the nanoparticles due to microcrystal parts or completely from an amorphous state?

Response: Thanks a lot for the valuable advice. The UV-vis absorption spectrum of water-dispersed TpPBr@F127 nanoparticles of TpPBr was measured and shown in the added Figure S21 below. This absorption spectrum was similar to that of TpPBr solid powder, whose absorption onset extends to the visible region at about 600 nm. Therefore, the nanoparticles can absorb light up to 600 nm, which thus can be excited at 488 nm. The selected area electron diffraction (SAED) images were collected for the nanoparticles, as the typical one shown in Fig. 5e. The image exhibited that partial nanoparticles were microcrystalline. Thus, the long-lived RTP (>200 ms) comes from the microcrystal parts. It is noted that the amorphous part of TpPBr nanoparticles also exhibited RTP, which can enhance the phosphorescent intensity of the nanoparticles. Thus, TpPX nanoparticles with efficient RTP can be simply prepared without overcaution for nanocrystal formation. The relative sentence has been modified to “Such a long-lived luminescence at a long-wavelength range of over 600 nm which can be excited by visible light over 450 nm was rarely reported for purely organic nanoparticles (Fig. 5f, Supplementary Fig. 21 and Table 8), which is an ideal reagent for bio-imaging.” in Lines 378-381 in the revised manuscript.

Added Figure S21. Normalized UV-vis absorption of TpPBr in THF (10^{-5} M), the aqueous solution of TpPBr@F127 nanoparticles and TpPBr powder.

7. Even though nanoparticle-based bioimaging was demonstrated, no afterglow imaging was shown; it is simple cellular imaging using 488 nm excitation and red channel emission. Afterglow bioimaging is required for the benefit of autofluorescence-free images.

Response: Thank the reviewer for the valuable suggestion. In order to show the benefit of afterglow for bioimaging, an in-vivo imaging experiment using TpPBr@F127 nanoparticles was performed. The nanoparticles were injected subcutaneously into mice and excited with a 405 nm flashlight for 30 s. After switching off the excited light, afterglow bioimaging photographs of living mice with autofluorescence-free were successfully captured by an IVIS instrument in bioluminescent mode (revised Fig. 5k). Furthermore, we could also capture a clear red afterglow using a mobile camera with video parameters of 1080p HD/60 fps, which significantly simplifies application scenarios by using an ordinary accessible mobile phone as a detector (revised Fig. 5j). These results show the broad prospects for biological applications of these organic afterglow nanoparticles, which can avoid the interference of spontaneous fluorescence in organisms.

The relative discussion about bioimaging has been added in Lines 383-391 in the revised manuscript as: “Additionally, as clearly shown in the images of Fig. 5i-k, the TpPBr nanoparticles were utilized for the imaging of RM-1 cells and living mice. A bright phosphorescence emission in the red channel was collected for the cells incubated with the nanoparticles. After illumination, afterglow bioimaging photographs with autofluorescence-free for living mice were successfully captured by an IVIS instrument in bioluminescent

mode or a camera with video parameters of 1080p HD/60 fps in an ordinary accessible mobile phone (Fig. j-k). These results demonstrate the broad prospects for biological applications of these organic nanoparticles, which can work without real-time external excitation and avoid the interference of spontaneous fluorescence in organisms.”

Revised Fig. 5i-k. **i**, Confocal microscopy images of RM-1 cells co-incubation with TpPBr@F127 nanoparticles. Cells were viewed in the blue channel for DAPI ($\lambda_{ex} = 405$ nm, $\lambda_{em} = 430 - 470$ nm), a red channel for TpPBr ($\lambda_{ex} = 488$ nm, $\lambda_{em} = 600 - 650$ nm), bright field and merged images, respectively. **j-k**, Visible-light excited afterglow in-vivo bioimaging photographs of living mice captured using (j) iPhone12 Camera before/after removing the flashlight and (k) an IVIS instrument in bioluminescent mode after removing the flashlight ($\lambda_{ex} = 405$ nm, 35 W/m²). TpPBr@F127 nanoparticles (with, green dashed circle) were injected into one upper hind leg while F127 nanoparticles without TpPBr (without, orange dashed circle) were injected into the other as a reference.

8. If the long-lived red emission of the crystals is mainly from the dimers of TpPBr, TpPF, and TpPMe, why does a native T1 emission band exist in the delayed spectra of crystals at RT?

Response: This should be due to some amorphous defects in the crystal surface due to the difficult to grow a perfect organic crystal without any defects. As shown in Figure R4a, the native phosphorescence of the TpPX monomer became more evident along the amorphous part increased in various states of crystal, amorphous powder, and highly concentrated solution.

Responsive Figure R4a. Normalized delayed luminescence spectra of TpPBr in a crystalline and amorphous state or its solution in 1,4-dioxane (10^{-2} M) in the air ($\lambda_{ex} = 365$ nm, delayed 8 ms).

Other minor points:

9. All the long-lived decay profiles show multi-exponential behaviors. Hence, it is important to analyze the different components at different emission wavelengths.

Response: Thanks for the reviewer's careful reading and comment. With respect to the time-resolved emission spectroscopy (TRES) of TpPX powder at the nanosecond scale measured at room temperature, the primary long-lived components, including T_1 , T_1^{dimer} , and T_2^{dimer} , all overlap. Therefore, the multi-exponential lifetimes at different wavelengths consist of these emissions together.

Responsive Figure R6. Normalized time-resolved emission spectroscopy of TpPX in crystalline state at nanosecond scale (298 K, air, $\lambda_{\text{ex}} = 405$ nm).

10. What is the effect of fluorescence lifetime because molecular packing via H or J-aggregation can affect the radiative decay time from singlet excited states?

Response: Thanks a lot for the comment. As discussed in Response 5 to Reviewer 1 above, the molecular aggregates of TpPX were further investigated through single crystal structure analyses using the molecular exciton theory. As the results shown in Table S7, TpP, and TpPF adopted purely J-aggregation in their crystals. Mixed H/J-aggregation formed in the single-crystal structures of TpPBr and TpPMe. The aggregates of TpPI and TpPOMe belong to J-aggregation in their crystals. However, the fluorescence lifetimes of these compounds summarized in Table S3 exhibited no strong correlation with the stacking mode of aggregation (H- or J- aggregation). This may be because the introduction of the halogen atom in TpPX significantly increases k_{isc} , leading to an obvious decrease in the fluorescence time.

11. Change Figure S6 y-axis legend to Normalized absorbance/ em. Intensity

Response: We thank the reviewer for pointing out this mistake. We have corrected the y-axis legend as “Normalized Absorbance/Em. Intensity” in Figure S6 below.

Revised Figure S6. Normalized UV-vis absorption (green), prompt luminescence spectra (purple dashed), and delayed luminescence spectra (pink) of TpPX in 2-Methyltetrahydrofuran (UV-vis: 5×10^{-5} M, 298 K, in air; Prompt: TpP and TpPOMe @ 10^{-3} M, other compound @ 5×10^{-5} M, 298 K, in air, $\lambda_{\text{ex}} = 310$ nm; Delayed: 1×10^{-5} M; 77 K, in air, $\lambda_{\text{ex}} = 365$ nm, delayed 8 ms). Prompt (blue dashed) and delayed (RT: red;

77 K: brown) luminescence spectra of TpPX crystalline powder (Prompt: 298 K, in air, $\lambda_{\text{ex}} = 365$ nm; Delayed: 298 K & 77 K, in air, $\lambda_{\text{ex}} = 365$ nm, delayed 8 ms). Sol., solution; Cryst., crystalline.

12. Significant absorption from 350-800 in Figure S6b is probably due to the negative absorption data normalized in Figure S6b.

Response: Thanks a lot. We have corrected it in the revised Figure S6 above.

13. Why is the peak centered at around 550 nm in Fig. S7 at 100K more intense compared to 77K?

Response: This is because the 550 nm (actual 560 nm) peak in Figure S8 at 100 K came from the high-lying triplet states of dimer aggregate (T_n^{dimer} is generally T_2^{dimer} , as discussed in Response 3 to reviewer 1 above).

Firstly, the delayed emission of powder at 500-560 nm (516 and 560 nm) only appeared at the low temperature of 77 K (Figure S8), which excluded its possibility of thermal activated delayed fluorescence (TADF). Moreover, as shown in the responsive Figure R1a, these delayed emission peaks at 516 and 560 nm were significantly different from either the prompt or delayed peaks of the isolated molecule (10^{-4} M, light-blue line) and the aggregates (10^{-2} M, blue line). This result excluded the possibility of TADF/triplet-triplet annihilation (TTA) and monomer phosphorescence (the first triple excited state (T_1), peaks at 480 and 520 nm) of these peaks. Therefore, the peaks at 516 and 560 nm may be the emission of aggregates, such as that from the high-lying triplet states of aggregates (T_n^{dimer} , generally T_2^{dimer}).

In order to clearly understand these peaks, we tested the delayed luminescence spectra of TpPBr in the crystalline and amorphous states in different atmospheres (in air/in a vacuum) at low temperatures. The results showed that delayed peaks at 516 and 560 nm only appeared in the condition of both at low temperature and in a vacuum, as shown in the responsive Figure R1a. These peaks could be observed both at 100 K and 77 K in a vacuum for the crystalline powder. However, it is noted that the amorphous powder mainly showed the emission peaks of T_1 (480 and 520 nm) at 77 K in a vacuum, while T_2^{dimer} peaks (516 and 560 nm) at 100 K (Figure R1b). Compared to the T_1 state of aggregates (T_1^{dimer}), the T_2^{dimer} has a larger energy gap for the transition back to the ground state (S_0). It is more significantly affected by non-radiative transitions and oxygen quenching. Therefore, significantly delayed emission from T_2^{dimer} can only be seen at low temperatures in a vacuum. When the temperature increased from 77 K to 100 K, the phosphorescence of T_2^{dimer} enhanced in crystalline powder or appeared in amorphous powder due to the thermal equilibrium

of excitons converting from T_1^{dimer} to T_2^{dimer} states. As the temperature further increased, the non-radiative transitions of T_2^{dimer} excitation were promoted, and its emission was weakened (reference such as *Mol. Phys.* **27**, 969-979 (1974); *J. Phys. Chem.* **91**, 819 (1987); *Chem. Rev.* **112**, 4541–4568 (2012)). Thus, the change for the band at 500-560 nm is irregular in the temperature-dependent emission spectra of Figure S8.

At low temperatures and in a vacuum, due to the efficient suppression of internal conversions and non-radiative transitions, a blue-shifted emission directly from a high-lying excited state could be observed in some systems. It has been reported in the literature that compounds containing halogen or carbonyl groups exhibited double phosphorescence at 77 K due to the large energy gap between excited states (such as 0.19 eV in *Nat. Commun.* **8**, 416 (2017) and 0.42 eV in *Angew. Chem. Int. Ed.* **61**, e202205556 (2022)). The energy gap between 515 nm and 600 nm was measured to be 0.34 eV, which makes it possible to observation of the emission from the T_2^{dimer} state. To verify this hypothesis further, spin-orbit coupling matrix elements were calculated at the time-dependent density functional theory (TDDFT) by Orca based on the M062X TZVP basis set, as shown in Figure R1c. The results showed that the energy gaps of T_2 - T_3 and T_4 - T_5 were 0.21 eV and 0.34 eV, respectively, which were two of the largest among the energy gaps between T_n s. The total spin-orbit coupling matrix elements of S_0 and T_3 - T_{10} can be up to 55.5 cm^{-1} , indicating that the radiative transition from T_n to S_0 was possible.

The time-resolved spectral decay of TpPBr crystalline powder in different atmospheres and temperatures under different conditions further confirmed this mechanism, as shown in responsive Figure R1d-i. The phosphorescent peak of T_1 was seen at the delayed time of ~ 50 ns, while the T_1^{dimer} peak could be observed a bit later at ~ 80 ns when TpPBr crystals were measured in the air at 298 K (Figure R1d and g, added into the revised Fig. 3g-h in the revised manuscript). If detected in the vacuum at 298 K, the T_1^{dimer} peak would appear earlier at 60 ns (Figure R1e). However, no peaks for T_2^{dimer} (typical peak at 560 nm) could be seen in these conditions. The significant emission peaks (560 nm) for T_2^{dimer} can only be observed after 50 ns at 77 K in a vacuum, which appeared at the same time and decayed along with the T_1^{dimer} peak at ~ 600 nm (Figure R1f). These results indicated that these peaks (T_2^{dimer}) were associated with the T_1^{dimer} peak, which further confirmed it as the emission from the high-lying excited state of aggregates.

Responsive Figure R1: a) Normalized prompt luminescence spectra of TpPBr in 1,4-dioxane and its crystalline powder measured in air at 298 K (upper, sol.: $\lambda_{\text{ex}} = 310$ nm; crys.: $\lambda_{\text{ex}} = 365$ nm); Delayed luminescence spectra of TpPBr in 1,4-dioxane in air and crystalline powder in a vacuum/in air at 298/77 K (lower, $\lambda_{\text{ex}} = 365$ nm, delayed 8 ms). b) Normalized delayed luminescence spectra of TpPBr in crystalline and amorphous state in a vacuum or air ($\lambda_{\text{ex}} = 365$ nm, delayed 8 ms). c) Spin-orbit coupling calculations of TpPBr-dimer based on the single crystal conformation. d-i) Normalized time-resolved emission

spectroscopy (TRES) of TpPBr crystalline powder in air/in a vacuum at 298/77 K at nanosecond scale ($\lambda_{\text{ex}} = 405 \text{ nm}$). (d, g) Air, 298 K; (e, h) vac., 298; (f, i) vac., 77 K.

14. *The comparative Table S1 and S6 must be reconstructed with lifetime, yield, and wavelength information.*

Response: Thanks for this professional suggestion. We have corrected it and modified Tables S1 and S8 in the revised SI.

15. *Correct 2nd d1 in Table S5.*

Response: Thanks for pointing out this mistake. We have corrected it in updated Table S7 in the revised SI.

Response to the Comments and Suggestions of Reviewer 3

We thank the reviewer for the appreciation of our work as “*Yang groups developed phenyl(triphenylen-2-yl)methanone based long and persistent RTP molecules, which glows on a second time scale in crystalline, amorphous states, water and heating conditions. The work is interesting...*”.

1. *Authors mentioned in the introduction section (line 67-69) that “delicate intermolecular interactions and oxygen barrier properties. Accordingly, some interesting RTP properties sensitive to mechanical force, oxygen, or water have been developed based on these delicate interaction [24-27].” If there are reports where scientists have already designed RTP molecules which are sensitive to mechanical force, oxygen and water, then what is the novelty of the present work?*

Response: As mentioned in our manuscript, some systems with RTP properties sensitive to mechanical force, oxygen, or water have been developed based on the delicate interactions in the crystal. However, to find a robust persistent RTP luminophore system whose phosphorescent emission is not restricted to the crystalline state or other rigid environments with cautious treatment is also highly demanded for practical applications, such as bioimaging. In this work, we report the first example of a single-component system (TpPX) with robust persistent RTP emission and afterglow in various aggregated forms, such as crystal, fine powder, and even amorphous states. Our experimental data reveal that the vigorous RTP emissions

rely on their tight dimers based on strong and large-overlap π - π interactions between polycyclic aromatic hydrocarbon (PAH) groups. The dimer structure can offer not only excitons in low energy levels for visible-light excited red long-lived RTP but also suppression of the nonradiative decays even in an amorphous state for good resistance of RTP to heat (up to 70 °C) or water. Furthermore, we demonstrate the water-dispersible TpPBr nanoparticle with persistent red RTP over 600 nm and a lifetime of 0.22 s for visible-light excited cellular and in-vivo imaging, prepared through the common microemulsion approach without overcaution for nanocrystal formation. As summarized in Table S8 in the revised SI, such a result is the rare organic deep red (over 600 nm) RTP system with a lifetime of over 0.1 s for in-vivo bioimaging excited by visible light. Therefore, the findings open the opportunity for developing single-component luminescent materials with robust phosphorescence in an amorphous state, through an effective strategy of forming tight dimer.

2. Authors showed and also discussed phosphorescence after-glow lifetime of up to second time-scale. However, it is not reflected in their decay profiles (showed in several places), where I can see the decay profile reaches flat after 20 ms.

I request authors should collect decay profiles in several 100 ms or in second time scale to confirm the after glow phenomenon in second time-scale. Lifetime decay profile is the ultimate proof whether it is glowing in the second time scale or not.

Also most of the delayed spectra reported here, where delay has been given only 8 ms. If it is glowing in the second time-scale, then the author should observe delayed spectra even in a 100 ms delayed case. Thus, I request authors should measure and report delayed emission spectra for at least one sample (TpPBr) collected in various delay time (ranging from 5 ms to 100 ms).

Response: We thank the reviewer for this suggestion. The figure mentioned by the reviewer should be Fig. 2h in the main text. This could be a misunderstanding caused by the unclear figure legend. Actually, it contains a double x-coordinate, including the upper one for the crystal powder lifetime that is up to seconds and the lower one in the range of milliseconds for the amorphous sample. This figure has been modified to avoid misunderstanding, as revised in Fig. 2h in the revised manuscript. The lifetimes (0.03 - 0.47 s) of crystalline powders of TpPX in the air have been summarized in the added Figure S7a in the revised SI, demonstrating that the TpPX compounds indeed have phosphorescent lifetimes up seconds. According to the reviewer's comments, the time-resolved delayed spectra of TpPBr powder at different gating times (5 -

99 ms) were also measured and shown in Figure S7b. It is noted that clear delayed peaks for TpPBr could still be observed after a delayed 99 ms. Moreover, the delayed spectra of the other TpPX compounds at a delay time of 99 ms were also collected and shown in Figure S7c. The relative sentence in Lines 133-134 in the revised manuscript has been modified to "...as shown in their delayed spectra measured at 8 ms (Fig. 2a) or even 99 ms (Supplementary Fig. 7) after turning off the excitation light."

Revised Fig. 2: Phosphorescent properties of typical dimer luminophore of TpPBr with robust RTP. ... h, PL intensity decay curves of crystalline and amorphous TpPBr powder in a vacuum or air at the peak of 610 nm (298 K, $\lambda_{ex} = 415$ nm). Crys., crystalline; Amor., amorphous.

Added Figure S7. a) PL intensity decay profiles of crystalline TpPX powder at their corresponding peaks (ca. 600 - 630 nm) (298 K, air, $\lambda_{ex} = 415$ nm). b) Delayed luminescence spectra of TpPBr powder with different delay times (298 K, air, $\lambda_{ex} = 405$ nm). Test slits were not adjusted. c) Delayed luminescence spectra of other TpPX powder with a delay time of 99 ms (298 K, air, $\lambda_{ex} = 405$ nm).

3. Although it is suggested that 77 K is not the good choice to measure the phosphorescence spectra, did the author try to measure in 77 K also?

Author measured the low temperature phosphorescence using dioxane as a solvent at 0 degree celcius. I guess dioxane is not transparent at 0 degree, as dioxane is not considered as glass freezing solvent. Authors should choose some glass forming solvent instead of dioxane.

Response: Actually, we measured the phosphorescence spectra at 77 K, using both 2-methyl tetrahydrofuran and 1,4-dioxane as solid solvents first. The measured delayed emission peaks of TpPBr in the two solvents matched each other perfectly (Figure R7). This indicated that 1,4-dioxane is a suitable solvent for the test despite its being non-transparent after solidification at low temperatures. However, as the concentration of the solution increased, the changes in the dimers' phosphorescence at ~600 nm were very small due to the overlap by the much highly emissive peak at ~540 nm at low temperature (Fig. 3f). So, we measured this changes in the solution of 1,4-dioxane at a much higher temperature of 0 °C, as 1,4-dioxane froze and became a solid at the temperature lower than 10 °C. Compared to 2-methyl tetrahydrofuran, 1,4-dioxane can completely be solidified at 0 °C, providing a rigid environment for the chromophore and minimizing the effect of temperature when testing its delayed luminescence spectra.

Responsive Figure R7. Delayed luminescence spectra of TpPBr in 2-methyl tetrahydrofuran or 1,4-dioxane (10^{-5} M, 77 K, $\lambda_{\text{ex}} = 365$ nm, delayed 8 ms).

4. “Under the same concentration of 10.0 mM, it was found that a new absorption peak clearly appeared at around 370 nm in the UV-visible absorption spectra of TpPBr in 1,4-dioxane solution (Fig. 3b). These results indicate that the TpPBr aggregated into dimers not excimers in a high concentration of 10.0 mM.”

“Thus, these results demonstrate that the red phosphorescent emission with the maximum peak at about 610 nm in the crystal and the amorphous TpPBr samples originated from the dimers.”

From the above mentioned studies in the aggregated states (high concentration in liquid, crystalline and amorphous), it is confirmed that phosphorescence emission is coming from the aggregated state but not from the monomeric state. But I did not understand why the authors think that it originated from dimeric state, not from trimer, tetramer and so on. Even in the amorphous condition, one cannot rule out higher aggregated states, as I am sure the particle dimension in the amorphous state is micron size. Also, in the aggregated particles in di-oxane there is no control over the number of particles forming aggregate even in the aggregates in dioxane (Authors can find out the dimension from DLS studies). Authors also showed strong pi-pi stacking interaction in the crystalline states, but it does not mean that dimers are only possible in the aggregated state.

Response: We would like to thank the reviewer for the careful reading and valuable comments. That the dimer is the main aggregate form in the aggregates was concluded from several experimental data. Taking TpPBr as an example, a green afterglow from isolated molecules with peaks at 470 and 505 nm was observed in a highly diluted solution (10^{-7} M) of TpPBr at 77 K. Upon increasing the concentration from 10^{-4} M to 10^{-3} M, the delayed emission red-shifted with peaks to 483 and 520 nm. Then, significant aggregates formed with peaks at 495 and 535 nm at a high concentration of 10^{-2} M (depicted in Fig. 3f). If this bathochromic shift was attributed to the formation of different aggregates, including dimer, trimer, tetramer, and so on, a mixed delayed emission of various aggregates should be observed. Nevertheless, only a sole phosphorescence at 400-550 nm was obtained in the highly concentrated solution, which was neither broad nor multiple mixed peaks. Therefore, the redshifts could be ascribed to monomer emission affected by the microenvironment changes, such as stronger intermolecular interactions between dimers and monomers. It was noticed that when the concentration rose to 10^{-2} M, the phosphorescence at 600 nm for the dimer emerged at these low-temperature spectra. With conversing the excitation wavelength from 365, 405 to 450 nm, the delayed emission peaks in the 550-650 nm range became more pronounced (Figure R8 a-b). It is well known that long-wavelength excitation is profitable for exciting low-energy aggregation states. Therefore, delayed emission in the 550-650 nm range should originate from the aggregates in these solutions. Under the excitation of 450 nm, the delayed spectra of TpPBr solution (10^{-3} M & 10^{-2} M) at low temperatures were almost identical, indicating the exact nature of the aggregates. Coincidentally, the low-temperature delayed spectrum of the amorphous powder, the highly concentrated solutions ($\geq 10^{-2}$ M), and

the crystalline powder were identical, implying that they are essentially the same in the nature of their phosphorescence (Figure R8 c). It is appreciable that dimer, the simple aggregation type, was the main aggregation form of TpPBr molecules in the aggregates in various conditions. Therefore, dimers should be responsible for the 550-650 nm phosphorescence emission based on the above evidence. Meanwhile, we tested the dynamic light scattering (DLS) of TpPBr in 1,4-dioxane solutions with different concentrations. There were almost no aggregates formed from the DLS results when the concentration was 10^{-3} M. Whereas, with a further increase of the concentration to 10^{-2} M, the DLS result indicated that the molecules started to form nano-aggregates with the dimension of ~ 300 nm (Figure R9).

The relative discussion “Additionally, a sole delayed luminescence at 400-550 nm was obtained in the highly concentrated solution at 77 K (Fig. 3f). This luminescence should belong to phosphorescence from monomers but not mixed multimers, as they were neither broad nor multiple mixed peaks along aggregates forming in the solution.” has been added in Lines 190-194 in the revised manuscript.

Fig. 3: Characterization of the formation of TpPBr dimer for its robust RTP. ...f, Delayed luminescence spectra of TpPBr in 1,4-dioxane solutions with incremental concentrations measured at 77 K ($\lambda_{\text{ex}} = 365$ nm, delayed 8 ms). ...

Responsive Figure R8. a-c) Delayed luminescence spectra of TpPBr (a-b) in 1,4-dioxane solutions with incremental concentrations measured at 77 K under different excitation at 405 and 450 nm, respectively. c) Normalized delayed luminescence spectra of TpPBr in 1,4-dioxane (10^{-2} M) and its crystalline and amorphous powder at 77 K ($\lambda_{\text{ex}} = 405$ nm, delayed 8 ms).

Responsive Figure R9. Dynamic light scattering of TpPBr in 1,4-dioxane solutions (10^{-2} M).

5. In line no. 269-275 authors mentioned “As revealed through single-crystal XRD of four compounds, including TpP, TpPF, TpPBr, TpPI, and TpPMe, their tight dimer structures were confirmed and ascribed as H- or J- aggregates by analyzing their molecular packing structures with the shortest distance of triphenylene plane in the range of 3.390 ~ 3.554 Å (Supplementary Table 4-5 and Supplementary Figs. 12”. I can see in all the cases the arrangement is head to tail fashion, so apparently it is J-aggregate type not H aggregate. However, it is better not to mention J or H aggregates unless there is a clear signature like shift

in absorption spectra etc. Surely, it cannot be H aggregate, as generally H aggregates are non-luminescent in nature.

Response: Thanks for the reviewer's suggestion. As also discussed in Response 5 to Reviewer 1 above, the stacking modes of their single crystals were carefully analyzed using the molecular exciton theory, and the results are summarized in the updated Table S7. As shown in Fig. 3d and S12, all TpPX compounds adapted a parallel stacking mode and formed dimers in their crystals. Among these compounds, TpPOMe and TpPI adopted a head-to-head parallel stacking mode, while other compounds (TpP, TpPF, TpPBr, and TpPMe) took a head-to-tail stacking mode.

In order to identify their exact aggregation type, the angles (θ) between the transition moment of the molecule and the interconnection of the molecular centers were calculated based on their single-crystal structures, as summarized in Table S7. The resulting angles for the dimers of TpP and TpPF were revealed to be 43.8° and 16.4° , respectively, which are lower than the critical angle (54.7°) for H-aggregation. So, the aggregates are purely J-aggregate for TpP and TpPF. In contrast, the two types of dimer in the crystals of TpPI and TpPOMe both belong to J-aggregate due to all the angles being above 54.7° . Regarding TpPBr and TpPMe, they also contain two types of dimers but with different angles above and below 54.7° simultaneously. They thus formed mixed H/J-aggregates in their single crystals. These results correspond to the experimental fluorescence properties of these TpPX compounds. For phosphorescence, the relationship became irregular partially due to the heavy-atom effect of bromo- and iodo- substituents.

The relative sentences, including “Correspondingly, two types of dimers of TpPBr could be formed between the molecular packing structures, which were analyzed to be H- and J- aggregation using the molecular exciton theory[49], respectively (Supplementary Fig. 11-12 and Table 7).” and “As revealed through single-crystal XRD of six TpPX compounds and Tp, their tight dimer structures were confirmed and ascribed as H- or J- aggregates by analyzing their molecular packing structures with the distance of triphenylene plane in the range of $3.374 \sim 3.695 \text{ \AA}$ ” have been modified in Lines 233-236 and Lines 283-286 in the revised manuscript. The relative discussion has also been added to the revised manuscript as: “The Tp core itself showed an H-aggregation in the crystal. After being substituted by the phenyl carbonyl unit, only TpPI and TpPOMe adopted H-aggregation in their two dimer types in the crystals. The dimers of TpP and TpPF belonged to J-aggregates. Regarding TpPBr and TpPMe, they contained two types of dimers and formed mixed H/J-aggregates in their crystals.” in Lines 287-291 in the revised manuscript.

Figure S11. Schematic representation of the aggregation models. The blue sphere and purple plane are the centroid and plane of the triphenylene core, respectively. And d_1 represents the vertical distance between two adjacent triphenylene planes, while d_2 is the centroid distance of two adjacent triphenylene planes. The red arrow describes the transition dipole moment of the monomer in the dimer. The green line is across the centroid of the monomer in the dimer. And the angle between the transition dipoles and the interconnected axis is indicated by θ . The transition dipole moment is calculated using the Gaussian 09W package.

Fig. 3d, The packing structure of TpPBr dimer in its crystal structure.

Figure S12. Crystal structures of TpPX. Distance represents the vertical distance between two adjacent triphenylene planes.

Added Table S7. The aggregation models of TpPX. Results are calculated based on their single crystal structures. Here, d_1 represents the vertical distance between two adjacent triphenylene planes, while d_2 is the centroid distance of two adjacent triphenylene planes. The angle between the transition dipoles and the interconnected axis is indicated by θ .

Compound-dimer	$d_1 / \text{Å}$	$d_2 / \text{Å}$	θ	transition electric dipole moments of S_0 - S_1 (Au)	H/J-aggregation
TpP	3.390	3.613	43.8	(0.1271, -0.0780, -0.0935)	J
TpPF	3.427	3.631	16.4	(-0.1434, -0.0841, 0.0960)	J
TpPBr	3.695	4.496	39.0	(-0.0498, -0.0752, -0.0994)	J
	3.554	3.823	76.9	(-0.0498, -0.0752, -0.0994)	H
TpPMe	3.590	4.823	34.3	(0.0414, 0.0642, 0.0954)	J
	3.472	3.748	68.7	(0.0414, 0.0642, 0.0954)	H
TpPI	3.399	4.483	66.4	(-0.0153, 0.0761, 0.1739)	H
	3.481	4.483	71.8	(0.0829, -0.0286, -0.0261)	H
TpPOMe	3.380	4.009	64.0	(-0.0829, 0.1141, -0.1253)	H
	3.480	4.564	66.0	(0.0736, -0.1177, -0.1160)	H
Tp	3.374	5.274	65.1	(0.0025, 0.0052, -0.0014)	H

Other modifications or deletions in Supporting Information files:

- (1) In the “2. Instrumentation and Test Methods” part of the revised SI, the detailed measured conditions of afterglow in-vivo bioimaging photographs are added. Some measured conditions have also been revised as the measurements changed.
- (2) The single crystal of TpPOMe was obtained recently, and its crystalline data was added to the updated Table S6 and Figure S12 in the revised SI. Additionally, the CCDC number of TpPOMe was also added to the “2. Instrumentation and Test Methods” part of the revised SI.
- (3) Some new calculated data have been added to the revised SI. The IRI of TpPBr-dimer, TpPMe-dimer2, and TpPOMe-dimers were added to Figure S13 and the NTO of TpPOMe was added to Figure S14.
- (4) The angles between the transition moment of the molecule and the interconnection of the centers were re-calculated and checked carefully according to their single crystals. The identification of H/J-aggregation in the TpPX crystal has been added to the updated Table S7 in the revised SI.

(5) Due to the weak fluorescence of TpPI and TpPOMe, the signal-to-noise ratio of their prompt luminescence spectra was very low when measured using an RF5301 fluorescence spectrometer. Therefore, prompt luminescence spectra of TpPX were replaced with the results collected from an Edinburgh FLS1000 steady/transient state fluorescence spectrometer to obtain their signals successfully. These new data have been added to the updated Figure S6 in the revised SI.

(6) The cryogenic phosphorescence of TpPBr powder in the original SI has been re-measured and updated in Figure S6.

(7) Afterglow video of amorphous TpPI powder at 298 K in air excited at 405 nm has been added to the revised SI as new Video S3.

REVIEWERS' COMMENTS

Reviewer #2 (Remarks to the Author):

The authors responded to reviewers' comments and revised the manuscript. The in vivo bioimaging included in the revised manuscript using red afterglow nanoparticles is important for background-free bioimaging using organic materials. Therefore, this revised manuscript is suitable for publication. However, there are some points that need to be corrected

1. As per Table S3, the experimental value of k_{isc} of Tp and TpP is comparable and indicates heavy atom play a more significant role here. Hence, authors should consider changing the introduction and discussion sentences accordingly "And the phenyl carbonyl unit is aimed to promote ISC efficiency, which has an electronic configuration of $3(n, n^*)$ with a fast ISC rate (Fig. 1b)[34]....."
2. The quantum yield of long-lived nanoparticles is useful information for the readers; therefore, authors should incorporate it in Table S8.
3. I am not convinced with the explanation of the irregular behavior of temperature-dependent emission of TpPBr. Why does the small change in temperature from 77K to 100K promote reverse IC compared to the fast IC process from T2 to T1? What is the nature of the T2 dimer, and why radiation from T2 and T1 is comparable only at 100K ($kT = 0.0086$ eV)? The explanation of radiation from Tn to S0 using the SOC is not appropriate. The authors stated that the k_p is not related to the SOC of Tn-S0 (eq 2, 3 in response file). The SOC of Tn-S0 is directly related to the non-radiative decay channels from the respective Tn levels. Therefore, the authors should address and incorporate these points in the manuscript or supporting file.

Reviewer #3 (Remarks to the Author):

I am satisfied with the revisions made by the authors, and therefore, I am recommending it for publication.

Response to the reviewers' comments and suggestions

Manuscript submitted to *Nature Communications*

Manuscript ID: NCOMMS-23-30644

Title: Visible-light-excited robust room temperature phosphorescence derived from a single-component dimer luminophore in the amorphous state

Authors: Danman Guo, Wen Wang, Kaimin Zhang, Jinzheng Chen, Yuyuan Wang, Tianyi Wang, Wangmeng Hou, Zhen Zhang, Huahua Huang, Zhenguo Chi, and Zhiyong Yang*

We sincerely thank the reviewers for their valuable comments. Corrections have been made according to the suggestions of the editors and the reviewer 2 (highlighted in yellow in revised manuscript) and are explained as follows:

Response to the Comments and Suggestions of Reviewer 2

1. As per Table S3, the experimental value of k_{ISC} of Tp and TpP is comparable and indicates heavy atom play a more significant role here. Hence, authors should consider changing the introduction and discussion sentences accordingly "And the phenyl carbonyl unit is aimed to promote ISC efficiency, which has an electronic configuration of $3(n, \pi^)$ with a fast ISC rate (Fig. 1b)[34]....."*

Response: Thank for the reviewer's valuable comments. However, the Q_{ISC} values of TpP and TpPMe are bigger than Tp, although they contain no heavy atom in their molecular structures. Additionally, TpPMe without heavy atom possessed comparable k_{ISC} and Q_{ISC} values to TpPBr. So, the phenyl carbonyl unit played a more important role on ISC improvement in these TpP derivatives. Therefore, the relative sentences have been kept the same in the revised manuscript.

2. The quantum yield of long-lived nanoparticles is useful information for the readers; therefore, authors should incorporate it in Table S8.

Response: Thank for pointing out this, the quantum yield (1%) of TpPBr@F127 nanoparticles in an aqueous solution was measured and added in Table S6.

3. I am not convinced with the explanation of the irregular behavior of temperature-dependent emission of TpPBr. Why does the small change in temperature from 77K to 100K promote reverse IC compared to the fast IC process from T2 to T1? What is the nature of the T2 dimer, and why radiation from T2 and T1 is comparable only at 100K ($kT = 0.0086$ eV)? The explanation of radiation from Tn to S0 using the SOC is not appropriate. The authors stated that the k_p is not related to the SOC of Tn-S0 (eq 2, 3 in response file). The SOC of Tn-S0 is directly related to the non-radiative decay channels from the respective Tn levels. Therefore, the authors should address and incorporate these points in the manuscript or supporting file.

Response: Thank you very much for your valuable comments. Firstly, as mentioned in our previous response (response 13), the experiment results have excluded the possibilities of TADF, TTA, and monomer phosphorescence for the emissions at 500-560 nm (peaks at 516 and 560 nm), which only appeared in the condition of both at low temperature and in a vacuum. So, the emission at 515 nm might be from the high-lying triplet states of aggregated states (T_2^{dimer}), due to the suppression of internal conversion (IC) and non-radiative transitions at low temperatures.

Compared to the T_1^{dimer} , the T_2^{dimer} has a larger energy gap for the transition back to the ground state (S_0). It is thus more significantly affected by non-radiative transitions and oxygen quenching. Additionally, the moderate energy gap (0.34 eV) between T_2^{dimer} and T_1^{dimer} made it possible to observe the emission from the T_2^{dimer} state. It is reported that double phosphorescence at 77 K was achieved due to the large energy gap between excited states. (such as 0.19 eV in *Nat. Commun.* **8**, 416 (2017) and 0.42 eV in *Angew. Chem. Int. Ed.* **61**, e202205556 (2022)). In the time-resolved spectral decay of TpPBr, the significant emission peaks (560 nm) for T_2^{dimer} appeared simultaneously and decayed along with the T_1^{dimer} peak at ~600 nm. However, the decay of T_2^{dimer} emission was much faster than T_1^{dimer} , indicating its short lifetime and subsequent faster radiation with bigger rate (k_p). Therefore, when the temperature increased from 77 K to 100 K, the phosphorescence enhancement of T_2^{dimer} was mainly relative to two reasons. One reason was that the thermal equilibrium of excitons converting from T_2^{dimer} to T_1^{dimer} states, as their moderate energy gap. Another reason was the bigger phosphorescence rate (k_p) of T_2^{dimer} , which could suppress the IC transition from T_2^{dimer} to T_1^{dimer} and subsequently promote the emission of T_2^{dimer} . As the temperature further increased,

the non-radiative transitions of T_2^{dimer} excitons were promoted, and its emission was weakened (reference such as *Mol. Phys.* **27**, 969-979 (1974); *J. Phys. Chem.* **91**, 819 (1987); *Chem. Rev.* **112**, 4541-4568 (2012)).

The relative discussions have been added to Supplementary Figure 8a in the Supplementary Information: “The emission at 515 and 560 nm might be from the high-lying triplet states of aggregated states (T_2^{dimer}), due to the suppression of internal conversion (IC) and non-radiative transitions at low temperatures. When the temperature increased from 77 K to 100 K, the phosphorescence enhancement of T_2^{dimer} was mainly relative to two reasons. One reason was that the thermal equilibrium of excitons converting from T_2^{dimer} to T_1^{dimer} states, as their moderate energy gap. Another reason was the bigger phosphorescence rate (k_p) of T_2^{dimer} , which could suppress the IC transition from T_2^{dimer} to T_1^{dimer} and subsequently promote the emission of T_2^{dimer} . As the temperature further increased, the non-radiative transitions of T_2^{dimer} excitons were promoted, and its emission was weakened (reference such as *Mol. Phys.* **27**, 969-979 (1974); *J. Phys. Chem.* **91**, 819 (1987); *Chem. Rev.* **112**, 4541-4568 (2012)).”

The relative discussions have been added to Supplementary Figure 10 in the Supplementary Information: “In the time-resolved spectral decay of TpPBr, the significant emission peaks (560 nm) for T_2^{dimer} appeared simultaneously and decayed along with the T_1^{dimer} peak at ~600 nm. However, the decay of T_2^{dimer} emission was much faster than T_1^{dimer} , indicating its short lifetime and subsequent faster radiation with bigger rate (k_p).”

Additionally, thanks to your suggestion, we have corrected the explanation of k_p and k_{nr} . k_p is approximately expressed as

$$k_p \propto \lambda_p^2 (\sum_m \mu_{s_m-s_0} \lambda_m)^2$$

$$\lambda_m \approx |\langle \psi_m^1 | \overline{H_{s_0}} | \psi_1^3 \rangle| / E_{s_m-T_1} \quad (m \geq 2)$$

And k_{nr} is often expressed based on a nonadiabatic SOC between T_1 and S_0 ($\langle \psi_1^3 | \overline{H_{s_0}} | \psi_0^1 \rangle$) as (*Chem. Phys. Lett.* **16**, 353-358 (1972)):

$$k_{nr} = \frac{2\pi}{\hbar} |\langle \psi_1^3 | \overline{H_{s_0}} | \psi_0^1 \rangle|^2 FC$$

Where FC is the Frank-Condon factor. According to the above equations, k_{nr} is directly related to the SOC of T_n - S_0 but not k_p . Therefore, the spin-orbit coupling (SOC) of S_0 and T_3 - T_{10} is incorrect for interpreting the radiative transition from T_n to S_0 in the first response letter.

The relative discussions have been added to Supplementary Figure 15 in the Supplementary Information: “Take TpPBr (b) as an example. The SOC matrix elements were calculated and showed that the energy gaps of T_2 - T_3 and T_4 - T_5 were 0.21 eV and 0.34 eV, respectively, which were two of the largest among the energy gaps between T_n s. The total spin-orbit coupling matrix elements of S_1/S_2 and T_3 - T_{10} were up to 55.5 cm^{-1} , indicating that the transition from S_1/S_2 to T_n (T_3 - T_{10}) was possible.”